# The Vascular Endothelium and Coagulation: Homeostasis, Disease, and Treatment, with a Focus on the Von Willebrand Factor and Factors VIII and V

**DOI:** 10.3390/ijms23158283

**Published:** 2022-07-27

**Authors:** Juan A. De Pablo-Moreno, Luis Javier Serrano, Luis Revuelta, María José Sánchez, Antonio Liras

**Affiliations:** 1Department of Genetics, Physiology and Microbiology, School of Biology, Complutense University, 28040 Madrid, Spain; jdepablo@ucm.es (J.A.D.P.-M.); luisserr@ucm.es (L.J.S.); 2Department of Physiology, School of Veterinary Medicine, Complutense University of Madrid, 28040 Madrid, Spain; lrevuelt@vet.ucm.es; 3Centro Andaluz de Biología del Desarrollo (CABD), Consejo Superior de Investigaciones Científicas (CSIC), Junta de Andalucía, Pablo de Olavide University, 41013 Sevilla, Spain; mjsansan@upo.es

**Keywords:** vascular endothelium, coagulation, embryo, von Willebrand factor, factor VIII, factor V, homeostasis, coagulopathies, treatment

## Abstract

The vascular endothelium has several important functions, including hemostasis. The homeostasis of hemostasis is based on a fine balance between procoagulant and anticoagulant proteins and between fibrinolytic and antifibrinolytic ones. Coagulopathies are characterized by a mutation-induced alteration of the function of certain coagulation factors or by a disturbed balance between the mechanisms responsible for regulating coagulation. Homeostatic therapies consist in replacement and nonreplacement treatments or in the administration of antifibrinolytic agents. Rebalancing products reestablish hemostasis by inhibiting natural anticoagulant pathways. These agents include monoclonal antibodies, such as concizumab and marstacimab, which target the tissue factor pathway inhibitor; interfering RNA therapies, such as fitusiran, which targets antithrombin III; and protease inhibitors, such as serpinPC, which targets active protein C. In cases of thrombophilia (deficiency of protein C, protein S, or factor V Leiden), treatment may consist in direct oral anticoagulants, replacement therapy (plasma or recombinant ADAMTS13) in cases of a congenital deficiency of ADAMTS13, or immunomodulators (prednisone) if the thrombophilia is autoimmune. Monoclonal-antibody-based anti-vWF immunotherapy (caplacizumab) is used in the context of severe thrombophilia, regardless of the cause of the disorder. In cases of disseminated intravascular coagulation, the treatment of choice consists in administration of antifibrinolytics, all-trans-retinoic acid, and recombinant soluble human thrombomodulin.

## 1. Structure, Physiology, and Function of the Vascular Endothelium

### 1.1. Overview

The endothelium is the innermost cellular layer of all blood vessels, from the largest arteries to the smallest capillaries and lymphatic vessels. Its main function is to serve as a barrier between the circulation and interstitial tissue [1]. Apart from this passive function, the endothelium also plays a secretory, synthetic, metabolic, and immunologic role. It regulates multiple physiological processes, such as vascular tone, angiogenesis, and membrane permeability [2,3].

Some pathological events, such as acute inflammation in the context of generalized sepsis or COVID-19 [4], tend to result in a cascade of inflammatory reactions potentially altering the vascular endothelium’s phenotype and giving rise to disruptions in the blood clotting process, higher permeability of the blood–brain barrier, and increased leukocyte chemotaxis toward certain locations [1,5,6,7]. Given its critical role in the inflammatory response, the production of proinflammatory mediators, and the efficient operation of the coagulation system and repair of damaged organs, the endothelium is a key factor in most stages of inflammatory disease and, consequently, a potential therapeutic target.

Although the endothelial cells that make up the vasculature of the different organs and species share common properties, they may present with different phenotypes, even across different vascular beds within the same organ [1]. This heterogeneity translates into changes in cell morphology and function, and in some genetic expression patterns [1]. Despite the complexity of this heterogenicity, endothelial cells have become therapeutic targets at the level of their specific membrane markers for a variety of conditions, such as cancer and pulmonary disease [5,6,8,9].

As a result of the foregoing, the vascular endothelium has ceased to be a mere spectator and has become a key player in a wide variety of functions. It is an endocrine organ that regulates blood flow and the exchange of fluids, nutrients, and metabolites and is a crucial mediator of several functions, such as the regulation of vascular tone through vasoconstriction or vascular relaxation, vascular remodeling, control of hemostasis and thrombosis, cell adhesion, smooth muscle cell proliferation, and vascular inflammation [10]. Breakthroughs, such as the discovery of prostacyclin and its endothelial origin [11,12], and a better understanding of the role of endothelial nitric oxide (NO) [13] have brought home the importance of the endothelium for vascular relaxation.

The supply of oxygen to bodily tissues depends on the synthesis and release of NO, while the balancing of vascular tone is contingent on numerous factors, such as the endothelium-derived hyperpolarizing factor, cyclooxygenase, lipoxygenase, and cytochrome P450, as well as on other molecules, such as angiotensin II, endothelin, urotensin, c-type natriuretic peptide, bradykinin, adrenomedullin, adenosine, purine, and the reactive oxygen species [14].

NO is a molecule that plays a pleiotropic role in endothelial function. Apart from its key role in the regulation of vascular tone through the relaxation of smooth muscle cells, NO also exerts an antithrombotic effect by attenuating platelet activation and aggregation [15], being also involved in maintaining the integrity and permeability of the endothelium [16].

### 1.2. Morphological and Functional Heterogeneity of the Endothelium

One of the most distinctive characteristics of the vascular endothelium, which also impacts its physiological function, is its heterogeneity across different organs [17]. During the development phase, endothelial cells originate from mesodermal precursors to form a primitive vasculature (vasculogenesis) [18,19], endothelial maturation occurring in response to epigenetic and environmental signals [20]. Regulation of vasculogenesis and angiogenesis takes place mainly as a result of the activity of the vascular endothelial growth factor (VEGF), angiopoietin, and the transforming growth factor [21]. This complex biogenesis gives rise to phenotypes characterized by highly specific spatial and temporal heterogeneity, associated with the different stimuli to which endothelial cells can be subject (functional heterogeneity) [22]. These characteristics are closely related with the differences observed across different biodynamic environments and the blood vessels’ heterogeneous functional properties [23].

The heterogeneity is also observed across different species [24,25,26,27,28], which is not without significance, as preclinical studies conducted to find potential treatments for different diseases are often based on animal models. This raises doubts on the concordance between the results of preclinical studies and those obtained in clinical trials [29,30].

### 1.3. Biomarkers of Endothelial Function

As an increasing body of knowledge develops on the properties of the vascular endothelium and increasingly more if its functions are uncovered, the need arises to come up with appropriate techniques and methods of studying such functions.

The use of plasma-based biomarkers to monitor endothelial function is associated with countless advantages, mainly related to the simplicity of the procedures involved [31]. These are applicable not only for diagnostic purposes but also for prognosis determination, even at the individual patient level [10,32].

Specific markers of endothelial activity are mainly vWF and thrombomodulin (TM), in both its soluble and conjugated forms. vWF is a marker of endothelial cell activation, while TM usually behaves as a marker of endothelial damage. Other markers, such as the ones described later, are not as specific for endothelial function but may become potential indirect markers of some alterations in the function of endothelial cells.

Given that endothelial activation is triggered by several inflammatory stimuli, the different inflammatory markers (C-reactive protein; CD40 ligands; interleukins, such as IL-1β and IL-18; chemokine ligand 2 [CCL2]; pentraxin-3; and sortilin) may be used to analyze the endothelial response [31]. Moreover, with regard to the activation of the endothelium by inflammatory stimuli, the cell adhesion molecules that are expressed (intercellular adhesion molecule-1, E-selectins, and vascular cell adhesion molecule-1) are considered valuable markers, particularly during the early stages of endothelial activation and systemic inflammation [33].

E-selectin is a type-C lectin that serves as one of the most specific markers of endothelial activation [34,35]. C-reactive protein, for its part, is also a marker but one that possesses proatherogenic properties that enhance the expression of adhesion molecules, reduce the bioavailability of NO, and induce endothelial dysfunction [36,37].

Other markers include the CD40 ligand, which is released by activated platelets and plays a key role in hemostasis and causes an inflammatory response in the vascular wall [38]. Another marker, CCL2, induces attraction of the monocytes from the vessel lumen into the subendothelial space and, together with adhesion molecules, promotes the binding of monocytes to endothelial cells [39]. Asymmetrical dimethylarginine, an endogenous inhibitor of NO synthase and a biomarker of NO-dependent endothelial dysfunction and atherosclerosis [40,41,42], is present at increased levels in patients with hyperlipidemia, hypertension, coronary artery disease, end-stage renal failure, myocardial infarction, stroke, and diabetes [31,43,44,45,46,47,48,49], as well as in those with COVID-19 [50]. Endocan (a soluble chondroitin/dermatan proteoglycan sulfate expressed and secreted mainly by the activated endothelium) plays a role in endothelial-dysfunction-derived atherosclerosis, promoting inflammation, cellular adhesion, and oxidative stress [51]. Myeloperoxidase, a member of the hemoperoxidase superfamily, is produced by activated neutrophils, monocytes, and tissue macrophages. It catalyzes the formation of reactive oxygen species (ROS) responsible for the oxidative damage suffered by lipids and proteins in the human body [33]. Myeloperoxidase is an effective biomarker of the onset and progression of atherosclerosis [52,53]. Lastly, pentraxin 3, which belongs to the same family as C-reactive protein, is a good predictor of functional recovery in patients undergoing heart surgery as it is involved in matrix remodeling [54].

A different category of markers is that of cellular biomarkers. Optimal functioning of the vascular endothelium is the result of a delicate balance between damage and repair. This process of homeostasis may be evaluated by an analysis of either mature endothelial cells liable to detach naturally or due to damage to the endothelium, or by the so-called cell-derived extracellular vesicles (EVs). Flow cytometry and fluorescence microscopy can be used to quantify this type of finding.

EVs are membrane vesicles sized between 0.1 and 1.0 μm [55] released by different types of cells, including the circulating endothelial cells themselves. They can thus be regarded as a new class of biomarkers of endothelial injury, associated with atherosclerosis and its related vascular complications (inflammation, thrombosis, and apoptosis).

High levels of EVs are usually present when endothelial cells are activated and undergo apoptosis; they are typically related to thrombogenesis and atherosclerotic plaque formation [56]. Other types of EVs, such as platelet EVs [57] and monocyte-derived EVs, [58] are also potential biomarkers of endothelial function. Platelet EVs have been associated with inflammation [59], blood clotting [60], thrombosis, and tumor progression [61].

### 1.4. The Endothelium and Coagulation

Maintenance of blood fluidity is, together with the prevention of thromboembolic events, one of the main functions of the vascular endothelium, which controls homeostasis through coagulation and fibrinolysis [62].

Pioneering studies published in the 1970s and 1980s [63] have already hypothesized that the strategic arrangement of endothelial cells on the vascular luminal surface places them in an ideal position to play a significant role in regulating the coagulation process. Believing that the vascular luminal surface could not be considered an inert surface that merely lined the vessel wall, the authors theorized that the endothelial surface contained molecules that could serve as sites for antithrombin III binding, TM synthesis, maintenance of low tissue factor levels, and prostacyclin generation. TM remains on the endothelial cell surface and binds to thrombin, which causes its coagulant activity to turn into anticoagulant activity. Subsequently, the complex is internalized, and TM is recycled on the endothelial cell surface, where it again interacts with thrombin [63]. The aforementioned studies have even suggested that activation of certain coagulation factors could play a “propagating” role.

The disruption of endothelial cells resulting from vascular damage at the level of the endothelium brings about the induction of tissue factor and the production of platelet activating factor and thromboxane, thus providing a model for a thrombotic state in which the endothelium is able to promote coagulation.

The vascular endothelium used to be considered a mere contact surface, which, on being damaged, released collagen fibers that induced platelet adhesion and aggregation [64]. It was a simple contact surface that induced coagulation. Recent findings on the functions of the vascular endothelium have led to a reassessment of the role of endothelial cells in the blood-clotting process, with new models being established to explain hemostasis [65,66]. The state of the art has evolved from a model where protein-based coagulation factors were believed to direct and control the process to one where coagulation is deemed to be regulated by the actual properties of cell surfaces and where specific cell receptors are considered to play a key role in the activation of coagulation factors.

Cells containing similar levels of phosphatidylserine may thus play different roles in hemostasis, depending on the surface receptors they harbor. It is therefore suggested that coagulation occurs not as “cascade” but as a process involving three superimposed layers: initiation, which takes place in a tissue-factor-bearing cell; amplification, where platelets and cofactors are activated to pave the way for large-scale thrombin generation; and propagation, where large amounts of thrombin are generated on the platelets’ surface. This cell-based model explains some aspects about hemostasis that a protein-centered model would be unable to account for [66].

## 2. Hemostasis: An Overview

The hemostatic system is a highly preserved machinery that prevents significant blood loss following vascular injury. It could be conceived as a series of cellular and biochemical mechanisms that work in a coordinated way to maintain the fluidity of circulating blood, form blood clots to prevent blood loss, and reestablish blood flow during the healing process [67,68]. Four elements are involved: the vascular endothelium, the platelet system, the coagulation cascade, and the fibrinolytic system.

Two dimensions of hemostasis may be considered: primary and secondary hemostasis [68,69,70]. The former is the process whereby a white or platelet thrombus is formed. Apart from forming the white thrombus, platelets play an important role in promoting blood clot formation and enriching existing blood clots. White thrombi are just one of the results of platelet activity but are extremely important in the formation of arterial blood clots. Platelets, the endothelium, and adhesion proteins are all involved in this process, which comprises activations, changes, and interactions between those different players and coagulation factor VII (FVII) [71,72,73]. Secondary hemostasis, also called the coagulation cascade, is a proenzymatic process whereby soluble fibrinogen is transformed into fibrin, which is initially soluble but is subsequently stabilized by activated FXIII (FXIIIa), also called fibrin-stabilizing factor. This insoluble fibrin forms a stable red clot (or thrombus) made up of fibrin fibers and different types of blood cells (leukocytes, platelets, and red blood cells) [73,74,75,76].

The coagulation cascade (Figure 1) consists of two interrelated pathways: extrinsic and intrinsic pathways. The former is triggered following a vascular injury, where the blood comes into contact with subendothelial cells. These cells express the so-called tissue factor (TF), which binds to inactive coagulation FVII and transforms it into activated FVII (FVIIa). The TF–FVIIa complex (also known as the initiator complex) in turn transforms factor IX (FIX) into activated FIX (FIXa) when TF levels are low and factor X (FX) into activated FX (FXa) when TF levels are high. Subsequently, FXa binds to activated factor V (FVa) within the phospholipids on the cell surface, bringing forth the next stages of the coagulation cascade.

Activation of the intrinsic coagulation pathway is a key stage in the coagulation process and contributes to the pathogenesis of arterial and venous thrombosis. Genetic studies on knockout mice have demonstrated the significance of some of the components of this pathway, such as FX, FXII, and prekallikrein (Pre-K). The use of FXIa and FXIIa inhibitors has been shown to reduce thrombosis at the expense of a slight increase in the number of hemorrhages compared to currently available anticoagulant drugs [77].

The intrinsic pathway [77] is initiated by contact activation, i.e., the onset of the coagulation cascade is induced by plasma coming into contact with any surface other than the vascular endothelium (such as the endothelial basement membrane or the connective tissue collagen fibers), adjacent to the area where the vascular damage has occurred. These collagen fibers provide a scaffold for the formation of the initiator complex, which requires the participation of high-molecular-weight kininogen (HMWK), Pre-K, and factor XII (FXII). In its inactive form, FXII possesses the necessary catalytic activity to convert Pre-K into kallikrein. The latter transforms FXII into activated FXII (FXIIa). Subsequently, FXIIa, in the presence of Ca^2+^, activates factor XI (FXI) and converts it into activated FXI (FXIa), which then activates FIX to FIXa, which, in turn, binds to activated factor VIII (FVIIIa). In the presence of calcium and phospholipids, FIXa and FVIIIa form the tenase complex (first amplification complex), whose main function is to activate FX. FXa then binds to FVa to form the prothrombinase complex, which converts prothrombin into thrombin in the presence of phospholipids and calcium.

Finally, thrombin activates fibrinogen and promotes its polymerization, giving rise to fibrin polymers, which are stabilized and insolubilized by FXIIIa. Once formed, these fibrin polymers interact with the platelet clot, which came into being during the primary hemostasis process, stabilizing it and plugging the vascular lesion. Thrombin plays a key role in the regulation of hemostasis as it interacts with multiple factors and components of blood plasma, apart from activating FXI, FXIII, FVIII, and FV, the last two factors being part of the tenase and prothrombinase amplification complexes, respectively. There are certain inhibitory proteins of the coagulation cascade that, at the same time, participate in the homeostasis of the cascade, such as activated protein C (APC), antithrombin, and the tissue factor pathway inhibitor (TFPI) [68,69].

The next process required to complete hemostasis, once the bleeding has stopped and the endothelium has regained its integrity, is fibrinolysis. Its purpose is to dissolve the clot that has formed by removing intravascular fibrin and restoring normal blood flow (Figure 2). Among other functions, fibrinolysis plays an important role in the regulation of cancer and metastasis, cell remodeling, neuroplasticity, fertility, etc. Fibrin clot solubilization is essential for successful healing. Thrombin itself activates this process by stimulating the synthesis and release of the tissue plasminogen activator (t-PA), by endothelial cells, and of urokinase (u-PA), which is responsible for mediating the transformation of plasminogen into plasmin. Plasminogen activator inhibitor-1 (PAI-1) is the main antagonist of this pathway. Moreover, transformation of plasminogen into plasmin is mediated by such coagulation factors as FXIa, FXIIa, and kallikrein. Apart from PAI-1, there are other molecules capable of inhibiting the transformation of plasminogen into plasmin, thus helping regulate fibrinolysis. The most important of these are α2-antiplasmin, α2-antimacroglobulin, and C1 esterase inhibitor. Plasmin in turn interacts with fibrinogen and fibrin, which causes its degradation. One of the main by-products of this degradation is the D-dimer, which is currently used as a marker of thrombosis in patients with COVID-19 [78,79].

### Hemostasis during the Neonatal Period

Given that hemostasis does not occur only during adulthood, it is essential to gain a thorough understanding of how hemostasis is achieved at the different stages of development. Indeed, the hemostatic system evolves and matures from conception to adult age, following a process known as *development hemostasis* [80]. Each component in the system evolves differently during the process, with both quantitative and qualitative differences being observed at different stages.

Maturation of the hemostasis system starts in the uterus [76,81,82,83,84,85,86]. It has been shown that coagulation and fibrinolysis can be detected in the fetus as early as the 10th day of pregnancy [76,87,88,89,90]. Both processes evolve throughout the neonatal period until adulthood, with significant quantitative and qualitative differences in the levels of the various proteins that participate in coagulation and fibrinolysis [82,86,91,92,93,94]. This variability, observed in neonates at different gestational stages [87,95,96], could be accounted for by differences in liver maturity or by the varying vitamin K concentrations observed in childbearing women [97,98]. All the components of the hemostatic system are present at childbirth but always at lower levels than at later developmental stages [80,81,84,86,87,99]. Differences may also be qualitative, i.e., the structure of some of the components may vary across different developmental stages and different ages [81,83,100,101,102]. Maturity of the different proteins is usually attained in the course of the first months of life, although some achieve their maturity during adolescence and others at adulthood [81,84,85,95,96,103].

Small-for-gestational-age full-term newborns present with significantly lower concentrations of several coagulation factors [104]. Premature babies also present with differences in the activity of several clotting factors, but their hemostatic system tends to be well balanced [97,105,106]. One of the main differences between fetuses/newborns and children/adults is the presence of protein S (PS) in free form (no complexes with C4bBP are present) [80].

The endothelium plays an important role in the hemostatic system given that—as mentioned above—hemostasis is not a simple cascade but rather the result of complex interactions at the level of the blood vessel walls, coagulation factors, and some kinds of cells, such as endothelial cells and platelets, as well as leukocytes and erythrocytes. It should therefore be easy to understand that there are quantitative and qualitative differences across the different components of hemostasis during development.

## 3. Homeostasis of Hemostasis

The term “homeostasis” was coined many decades ago [107,108] to denote a variety of mechanisms that control the stability of the internal environment in terms of its (chemical and cellular) composition and of its physical and chemical variables. Although the concept of homeostasis has been called into question following a series of significant findings [109], when referring to hemostasis it could be said that homeostasis denotes the mechanisms aimed at ensuring that blood maintains its biological and physical-chemical properties and that it flows through the circulatory system to perform its different functions. In short, homeostasis is about striking a fine balance between the coagulation system, which prevents hemorrhages and blood loss, and the fibrinolytic system, which prevents thromboembolism.

Nevertheless, both coagulation and fibrinolysis [110] have their own homeostasis mechanisms and the proteins responsible for maintaining them ensure the regulation of homeostatic turnover at levels appropriate to each component. Excessive activation of the coagulation cascade may result in thrombotic processes, such as venous thromboembolism, which may lead to deep venous thrombosis and pulmonary embolism [77,111,112].

Generally speaking, the proteins that participate in the coagulation cascade and in the fibrinolytic process are protease enzymes (many of them serine proteases) that activate one another by proteolytic cleavage in a predetermined order and as a function of other activating factors, including (pro)thrombin, FVIIa, FIXa, FXa, and FXIa, and other serine protease cofactors, such as TM, TF, FVa, and FVIIIa. In addition, many inhibitory proteins, such as APC, a natural anticoagulant, are also serine proteases themselves. The balance, i.e., homeostasis of the coagulation-fibrinolysis system, is based on a delicate balance between proteases. In addition, the negative control of coagulation and fibrinolysis is exercised through protease inhibitors, such as serin protease inhibitors (SERPINs) [113]. Anticoagulant SERPINs include antithrombin, heparin cofactor II, protein Z-dependent protease inhibitors (ZPI), protease nexin 1, and the C1-inhibitor. The inhibitory activity of SERPINs is complemented by the action of a numerous group of non-SERPIN anticoagulants, such as TFPI, which is the primary inhibitor of the TF-FVIIa complex [114]. In turn, APC, together with PS, its main cofactor, is a crucial physiological inhibitor of FVa and FVIIIa [115]. The complex formed by TM and thrombin activates PC, resulting in higher levels of clotting efficiency when the endothelial cell protein C receptor (EPCR) binds to PC’s Gla domain. TM contains a series of domains, known as endothelial growth factor (EGF)-like repeats, which are involved in PC activation [110].

This striking kind of homeostasis, which has been called *proteostasis* [116], is based on a dispute between activating and inhibitory proteases (including protease inhibitors), which avail themselves of sophisticated defensive mechanisms, such as the ubiquitination–proteasome system [117]. This dispute is also common in other processes, such as those involved in aging or in the etiopathogenesis of neurodegenerative conditions [118].

Recent findings have demonstrated that the function of organelles and cellular responses to stress is crucial for determining vulnerability and resistance to environmental stress. At the molecular level, maintenance of proteostasis is a key mechanism for adapting to cellular stress. Disturbances in proteostasis at the level of the endoplasmic reticulum, which is a major site for protein folding within the cell, could be attributable to cellular stress and therefore be connected to the etiopathogenesis of certain diseases [116].

Until a few decades ago, the role of proteases was thought to be limited basically to controlling the turnover of certain proteins. Nowadays, however, thanks to the vast amount of knowledge generated around them, it is known that proteases are associated with the onset and progression of certain conditions, many of them age related, and that they are crucial for health. They contribute to the adaptive systemic signaling of the stress induced in the skeletal muscle proteasome [119], to intracellular signaling [118], and to the onset and progression of certain neurodegenerative conditions, such as Parkinson’s [120], Huntington’s [121], and Alzheimer’s [122] diseases. Surprisingly, it may even be the case that depending on the proteases at play, one same molecule or coagulation factor, such as FV, could perform a procoagulant or an anticoagulant function [123,124,125,126].

Of particular interest is the proteostasis observed in coagulation and fibrinolysis systems, which is associated with conditions such as congenital and acquired coagulopathies, thromboembolic or cardiovascular events, or dysfunctions arising following anticoagulation procedures.

Many (existing or under-development) molecular pharmacology protocols emphasize the need to target certain proteases in order to alter or restore homeostasis (proteostasis). Current treatment of certain congenital coagulopathies is based on the use of extended half-life exogenous recombinant coagulation factors [127] capable of neutralizing the action of matrix-degrading proteases. They are also based on the activation and inactivation of the proteases involved in the maturation and activation of, for example, transforming growth factor, myostatin, complement C3 and C5, or IL-1b and IL-18 [128], and on the inactivation of the retroviral proteases of HIV [129] or SARS-CoV-2, both at the level of the intracellular life cycle of the virus (protease 3CL [130,131]) and by facilitating the entry of the virus into the cell (TMPRSS2 [132]).

## 4. Molecular Structure and Clotting Function

The molecules’ structure determines their function. In the case of the proteins involved in the coagulation cascade, this concept becomes particularly important as the body’s response to bleeding must be swift and effective. There are a series of structures in nature that have been meticulously designed to fulfil certain highly important functions where error is not an option. Such is the case of the blood coagulation process, where exponential signal amplification can only be achieved through the activation of zymogens, a series of non-functional pre-structured (precursor) proteins that generally perform protease-like functions when they become activated [133].

Under physiological conditions, activation of a zymogen occurs as a result of an irreversible catalytic cleavage of one or several peptide bonds following limited proteolysis. It is a control mechanism that differs from other reactions, such as allosteric transitions or reversible covalent modifications [134].

The coexistence of several readily activated precursor zymogens makes available to the coagulation cascade the optimal amplification of a stimulus, such as an injury to the vascular endothelium, and allows a fast and exponentially more effective repair response, which has also been shown to have feedback potential [135].

The structure–function relationship thus becomes highly significant because activation of a certain coagulation factors following proteolytic cleavage can only occur if the said factor possesses within its structure sites that are sensitive and specific to protease activity. However, such coagulation factors must harbor other sequences that are capable of interacting with the molecules responsible for modulating the activation process.

It is generally understood that endothelial cells not only participate in the coagulation process but are also involved in the synthesis of von Willebrand factor (vWF) and FVIII. They also play an indirect yet crucial role in supporting the activation and inactivation of some coagulation factors, such as FVIII itself and FV. It could be said that all three factors (vWF, FVIII, and FV) share a close relationship with the vascular endothelium, participating in its biosynthesis, its activation, and its inactivation. The homeostasis of coagulation and fibrinolysis is based, precisely, on maintaining an appropriate balance between these three processes.

### 4.1. Molecular Structure of von Willebrand Factor

vWF (Figure 3) is a multimeric protein that is part of the subendothelial matrix of endothelial cells, where it is secreted by the Weibel–Palade bodies [136]. During primary hemostasis, it binds to the platelets and circulates in the bloodstream, where it forms a complex with (and stabilizes) FVIII [137,138,139].

The gene that codes for vWF is located in region p13.2 of chromosome 12 (12p13.2) and consists of a total of 52 exons, measuring around 175 kb in total [137]. This factor is synthetized mainly in the endoplasmic reticulum of endothelial cells and megakaryocytes as a pre-protein formed by a signaling peptide, a propeptide, and a mature protein monomer comprising 2813 amino acids [140,141]. These molecules subsequently undergo glycosylation and sulfation, carboxy terminal end dimerization, amino terminal multimerization, and proteolytic cleavage. Mature vWF and its propeptide are stored in the above-mentioned Weibel–Palade bodies and in the platelets’ alpha granules [136,137,140]. Some evidence indicates that FVIII is also stored in these locations [140].

The mature vWF protein is made up of different domains: a D3 domain, which binds to FVIII; an A1 domain, which binds to glycoprotein Ib alpha (GPIba) and to collagen VI and IV; an A2 domain, which binds vWF to ADAMTS13, which is responsible for degrading vWF; an A3 domain, which binds to collagen I and III and acts as a bridge through which vWF binds to the subendothelial collagen in the area where the vascular damage has occurred; a D4 domain; six C domains, where C4 binds to the activated platelets’ GPIIb-IIIa receptor; and a cystine-knot (CK) domain [140,141,142].

When immature vWF (pre-pro-protein) reaches the Golgi complex, a series of prodimers gather to form a dimeric bouquet, aligning themselves side by side. During the multimerization process, prodimers organize themselves into a right-handed helical structure, forming the wall of a hollow tube. vWF tubules assemble into so-called ministacks, which represent the initial structure of the Weibel–Palade bodies, also known as small Weibel–Palade bodies. As vWF matures, most of it is secreted into the plasma, and the majority of high-molecular-weight multimers assemble into larger Weibel–Palade bodies [62,140,143].

Following the synthesis and storage of vWF in the endothelial cells’ Weibel–Palade bodies, its secretion occurs by exocytosis. In the event of an increase in intracellular pH, the structure of vWF collapses and the factor ceases to be secreted by the Weibel–Palade bodies [140]. When certain agonists are used to stimulate the endothelium [144], vWF secretion is massive, with vWF multimers assembling into large vWF chains capable of recruiting large amounts of platelets and promoting their aggregation [140,143,145]. vWF degradation is mediated mainly by ADAMTS13, a metaloproteinase that cleaves tyrosine 1605 from methionine 1606 in the A2 domain [146].

### 4.2. Molecular Structure of Factor VIII

The gene that encodes FVIII (*F8* gene) is located in region q28 of the long arm of chromosome X (Xq28). It is a large gene measuring around 186 kb comprising 26 exons and 25 introns, exon 14 being the largest of all. FVIII’s mature messenger RNA, which encodes a glycoprotein containing around 2332 amino acids, is about 7 kb in size. In its inactive form, FVIII has six domains (Figure 4): A1, A2, B, A3, C1, and C2. The A1, A2, and B domains constitute the heavy chain, which weighs approximately 92–200 kDa [147], and the A3, C1, and C2 domains form the light chain, which weighs around 80 kDa. When inactive, the weight of this protein is around 293 kDa [147,148,149,150].

The active form of FVIII (FVIIIa) is an intrinsic pathway procoagulant protein that binds to FIX to form the tenase complex, which mediates the activation of FX to FXa. To perform this function, FVIII needs to be activated to FVIIIa by mediation of thrombin and/or FXa. Thrombin binds to residues Arg372 and Arg740 of the heavy chain and to residue Arg1689 of the light chain, which results in the protein being cleaved. As a result of this process, the B domain is obliterated and the resulting FVIIIa consists of a trimeric protein formed by an A1 domain (amino acids 1–372); an A2 domain (amino acids 373–740), both forming the heavy chain (with a weight of 92 kDa); and the A3, C1, and C2 domains (amino acids 1690–2332), which form the light chain (with a weight of 80 kDa). Following activation of FVIII, the FIXa-binding sites required for the formation of the tenase complex and the activation of FX become exposed [149,151].

FVIIIa binds to FIXa through mediation of amino acid residues 336, 558–565, 1810–1818, and 1719, giving rise to the tenase complex and to FX through residues 361–363, 400–409, 2007–2016, and 2253–2270 [148,150]. Binding to vWF is mediated by the A3 domain, with sulfation of Tyr1680. The C1 and C2 domains and platelet membrane phospholipids play an important role in this process. The C2 domain exerts its function through residues Met2199/Phe2200 and Leu2251/Leu2252, which are the ones involved in binding to vWF [151,152].

Proteolysis of FVIII occurs through mediation of APC, which binds to residues Arg336 and Arg562, cleaving the A2 domain and therefore inactivating the function of FVIIIa [126,153]. This inactivation can also be carried out by FXa by binding to the same residues as APC and causing the same cleavage and inactivation [154].

### 4.3. Molecular Structure of Factor V

In humans, the FV gene (*F5*) is located in the q23 region of the long arm of chromosome 1 (1q23). It is a large gene, measuring around 80 kb and comprising 25 exons and 24 introns, exon 13 being the largest one. The gene’s mature messenger RNA, which encodes a protein of around 2224 amino acids, measures around 6.8 kb [124,155,156,157]. In its inactive form, FV has six domains (Figure 5): A1, A2, B, A3, C1, and C2. Domains A1 and A2 constitute the heavy chain; the B domain, entirely encoded by exon 13, makes up the posttranslational region; and domains A3, C1, and C2 form FV’s light chain. The weight of inactive FV is around 330 kDa [157,158,159]. 

To perform its procoagulant function, FV is activated to FVa by mediation of thrombin, which acts at the level of FV’s B domain, binding to amino acids Arg709, Arg1018, and Arg1545. The FV protein is cleaved at these sites and undergoes changes in its constitution and structure, which facilitates exposure of the domains where the binding of FV to FXa usually takes place. The cleaved portion, which constitutes almost all of the B domain, results in the generation of FVa. The newly activated factor is made up of two chains: a heavy chain comprising the A1 and A2 domains and weighing 105 kDa and a light chain formed by the A3, C1, and C2 domains, weighing 71–74 kDa. FV can also be activated by FXa [124,158,160,161]. FVa binds to FXa to form the prothrombinase complex, which converts prothrombin into thrombin [157,158].

The C1 and C2 domains of FVa (amino acids Y1956 and L1957 for the C1 domain and W2063, W2064, and L2116 for the C2 domain) play a role in the binding of FVa to platelet membrane phospholipids, specifically to the phosphatidylserine residues [157,158]. The A1 and A2 domains of the heavy chain and the A3 domain of the light chain are responsible for the binding of FVa to FXa to form the prothrombinase complex. FVa binds to FXa at the sites of amino acid residues 311–325, 323–331, 400, 499–506 [158,160], and 510–709 [157,158]. Binding at residues 323, 324, 330, and 331 has been shown to be essential for the successful conversion of prothrombin into thrombin [157,162]. At the same time, FXa binds to FVa through residues 273 and 420, part of the region between residues 263–274, part of the 415–428 helix, and the 344–352 segment [163], while it is the Arg271 and Arg320 residues of FXa in the prothrombinase complex that mediate the conversion of prothrombin into thrombin [162].

Homeostatic regulation of coagulation, aimed at avoiding a state of hypercoagulability, is the responsibility of APC, a natural anticoagulant. C-protein is activated by thrombin, which is generated through the coagulation cascade and retro-inhibits the whole process. Thrombin, together with TM, interacts with the vascular endothelium to produce APC [164,165,166]. Inactivation of FVa by APC requires the participation of PS as a cofactor. The binding point is the endothelial cell or platelet membrane where the APC–PS–FVa complex is formed. APC subsequently binds to FVa’s Arg306, Arg506, and Arg679 residues, causing the cleavage of the A2 domain, which leads to FVa inactivation and to the discontinuation of its activity as a cofactor of the prothrombinase complex [157,161,167].

The FV molecule is exceptional in that, apart from its procoagulant function, it is endowed with an anticoagulant function, which it performs in two ways: one of them through inactivation of the tenase complex and the other at the level of the initiation of the tissue factor pathway [125,126,161,168]. With regard to the tenase complex inactivation pathway, APC binds to FVa’s Arg306, Arg506, and Arg679 residues and, together with PS, brings about the cleavage of the A2 domain from the tenase complex’s FVIIIa (FVIIIa–FIXa), which results in the inactivation of FVIIIa. An important role in this process is played by the C-terminal end of the B domain (last 70 amino acids) [124] of FV, which boosts the binding affinity of the APC–PS–FV complex for FVIIIa, which is crucial for successful inactivation. When FV is activated and the B domain is obliterated, the factor loses its function as an anticoagulant molecule [161]. With respect to its anticoagulant function at the level of the tissue factor pathway, FV may bind to the complex formed by TFPI and PS, increasing the anticoagulant efficacy of TFPI [126].

TFPI is an anticoagulant protein produced mainly by endothelial cells. Around 90% of TFPI circulates freely in plasma bound to lipoproteins, and the remaining 10% is stored in platelets and gets released during the platelet activation process. TFPI has three functional domains; one of them binds to FXa, leading to a conformational change and making the second domain bind to the TF–FVIIa initiator complex. The third domain is bound to transport lipoproteins. The result is the formation of an inactive four-element complex (TFPI–FXa–FT–FVIIa) [126,169]. In this way, TFPI and PS bind to FV at the C-terminal region, just before the area where the binding (and activation) of thrombin takes place (Arg1545), forming a TFPI–PS–FV complex and enhancing TFPI’s anticoagulant activity. Although neither FV nor PS are essential cofactors for TFPI’s anticoagulant function, they do increase the protein’s efficacy, eightfold in the case of PS and twofold in the case of the binding of FV to the TFPI–PS complex [126,161].

## 5. Clotting Factor Biosynthesis

Many of the mechanisms through which hemostasis participates in the development of embryos as compared with that of adults remain unknown. What is well known is that it is involved in the establishment and maintenance of vascular integrity during embryonic development, remodeling the embryonic blood vessels by angioblast differentiation, vessel alignment into vascular cords to form a primary vascular plexus, and the subsequent generation of further endothelial cells [170,171].

Neonate levels of the majority of coagulation factors generally amount to approximately 50% of those adults, with most individuals reaching adult levels within the first 6 months of life. Others do not reach those levels until adolescence [80].

### 5.1. Von Willebrand Factor

von Willebrand factor (vWF) levels are typically elevated at the time of birth [83,172], and usually decrease between the 3rd and 12th year of life [76,84]. In addition, neonates exhibit high levels of large vWF multimers [76,83,100,103,172,173], which is associated with more effective platelet aggregation and firmer adhesion to the vessel wall [83,174,175]. This may be a way of compensating for the relatively lower platelet function observed in neonates [176].

During embryonic development, Coffin et al. [177] reported that no vWF is produced until developmental stage E8. These authors observed that the first tissues to produce vWF were those of the dorsal aortae, the aortic arches, the intersomitic arteries, and the cardinal veins, with no vWF being detected in angioblasts. From gestational week 20, vWF levels, like those of other coagulation factors, tend to normalize. Moreover, a stronger binding of vWF to collagen has been observed [173,178,179]. Some authors have argued that the distribution of vWF multimers in neonates depends on the existence of a fine balance between the synthesis and secretion of vWF and the size of the vWF protein, which is regulated by ADAMTS-13 [180]. Elevated vWF levels are beneficial against von Willebrand’s disease in neonates as they help prevent cranial bleeding during childbirth [85].

### 5.2. Factor VIII

At birth, FVIII levels are usually normal or elevated [87,95,181,182], adult levels being typically reached at 6 months [97].

FVIII is potentially associated with angiogenesis during the intermediate stage of embryonic development, although FVIII-knockout (KO) mouse embryos typically display uneventful embryonic development despite experiencing postnatal bleeds [183]. Even though not much is known about the exact role of FVIII in neoangiogenesis during embryonic development, work on adult mouse models with hemophilia A has revealed the formation of large and irregular vessels, with high concentrations of vascular remodeling markers, such as αSMA, endoglin, and VEGF, in addition to the formation of an abnormal vasculature following induction of hemarthrosis [184]. FVIII has also been associated with decreased endothelial cell adhesion, which could lead to higher rates of cell migration in the context of angiogenic sprouting and changes in the expression profile of genes originating from endothelial cells associated with the functional pathways involved in neural tube morphogenesis, cellular adhesion/migration, and the immune response. FVIII has also been associated with enhanced in vitro vascular tubule network formation and increased vascular endothelial permeability, among other effects [185].

While FVIII production by endothelial cells has been addressed by many authors, few studies have focused on identifying the FVIII-producing loci during development. Our group has described the production of *F8* gene mRNA in different embryonic/fetal tissues, identifying a predominant window of expression at day 12 of embryonic development (E12) in the aorta/gonad/mesonephros (AGM) region and the liver. Moreover, an analysis of isolated cells from the fetal liver showed that VE-cad^+^ CD45^-^ endothelial cells resulted in higher levels of *F8* gene expression compared to hematopoietic (CD45^+^) cells or hepatocytes (Dlk1^+^) [186], supporting the notion that FVIII is produced in embryonic endothelial cells in similar quantities as in adults. Recent studies using genetically engineered mice in which the enhanced green fluorescent protein (EGFP) rather than the *F8* gene transcript was expressed under the regulation of the F8 locus showed that from developmental stage E12, liver sinusoidal epithelial cells (LSECs) exhibit varying amounts of EGFP, FVIII-producing LSECs being visible from the early stages of development. According to these authors, FVIII production becomes a feature of LSEC maturation as levels of LSEC EGFP increase with liver development [187].

Experiments carried out over three decades ago showed that liver transplantation cures hemophilia [188]. Consequently, initial studies seeking to establish whether endothelial cells are able to produce FVIII have focused on LSECs. Supportive evidence came from transplantation experiments showing that LSECs originating from wild-type mice engraft the liver sinusoids and correct the abnormal phenotype in mice mutants to restore FVIII production [189]. Further studies in humans showed that cultured primary microvascular endothelial cells from the lung, heart, gut, and skin, but not from umbilical vessels, constitutively secrete FVIII, suggesting that endothelial cells other than LSECs could potentially contribute to the FVIII pool [190].

To avoid the potential bias resulting from the modifications of the cell phenotype that can occur during in vitro culture, human Tie2^+^CD32b^+^ LSECs were separated from hepatocytes and CD11b^+^ macrophages by fluorescence-activated cell sorting (FACS) and FVIII activity was directly measured in cell extracts. This approach showed that LSECs, but not hepatocytes, contain FVIII [191]. Studies using in vivo genetic approaches confirmed that FVIII production is mostly restricted to endothelial cells. Transgenic mice were developed for conditionally blocking FVIII synthesis by endothelial cells using the Tie2-Cre x Lman^fl/fl^ genetic system. The procedure resulted in a drastic reduction in FVIII plasma levels [192].

A similar strategy was used to deplete the *F8* gene in an endothelial-cell-specific manner by crossing *F8* gene^fl/fl^ mice with various tissue-specific Cre strains, including endothelial cell drivers Cdh5 and Tie2, hematopoietic driver VAV, and hepatocyte driver Alb. The experiment confirmed that efficient endothelial KO models displayed a severe hemophilic phenotype with no detectable FVIII plasma levels [193]. As depletion of FVIII from endothelial cells using generic genetic drivers did not clarify the type of endothelial cells responsible for FVIII production, further *F8* gene mRNA transcriptomic profiling assays were performed on subsets of FACS-isolated endothelial cells from adult mice [194]. Endothelial cells expressing *F8* gene mRNA included LSECs, renal glomerular endothelial cells, and lymphatic endothelial cells, with elevated levels of transcripts from HEV-epithelial cells, which specialize in recruiting lymphocytes from the circulation. No *F8* gene mRNA expression was detected in endothelial cells originating from the microvasculature of any of the other tissues.

Pan et al. also showed that biologically active FVIII is present in lysates and supernatants of cultured purified lymph node epithelial cells (LECs) and LSECs but not in those of conventional blood capillary epithelial cells [194]. In line with these results from mouse models, a study by Zanolini et al. where different tissue sections were stained showed that FVIII expression is predominant in the postcapillary venules of human lymph nodes compared to the small arteries in the spleen [195]. The tissue analyses in that study showed that while FVIII is produced by endothelial cells in venous vessels, it is virtually undetectable in arterial endothelial cells, suggesting that the high blood flow characteristic of arteries may play a role in down-regulating FVIII expression. Altogether, these studies revealed localized endothelial cell expression of FVIII in the steady state and identified endothelial cells from liver sinusoids, glomeruli, and lymphoid vessels as major cellular sources of FVIII. This heterogeneity of endothelial cells can have important implications for FVIII biosynthesis as, according to previous reports, the co-expression of FVIII and vWF in vitro may result in increased stable accumulation of both factors in the Weibel–Palade bodies [196,197].

Since endothelial cells are the primary source of FVIII in the body, a number of studies have focused on the capability of endothelial progenitor cells (EPCs) to produce FVIII, a matter of great interest for the potential applicability of EPCs in cell therapy [198]. EPCs can travel in the circulation or reside in the organs’ vasculature. In the circulation, EPCs have been classified into two groups: myeloid angiogenic cells (MACs), of hematopoietic origin, and endothelial colony-forming cells (ECFCs), a type of in vitro outgrowth of circulating endothelial cells with high proliferative potential. Different studies have investigated the potential use of lentiviral-FVIII-transduced ECFCs in cell-based therapy to correct FVIII deficiency [199,200]. Although primary umbilical cord blood ECPCs present with almost undetectable levels of FVIII [201], cytoplasmic staining has shown that ECPCs from adult peripheral blood could express FVIII [202]. With regard to tissue-resident EPCs, pre-cultured human CD31^+^c-Kit^+^VEGFR^+^ cells isolated form umbilical cord tissue (UCT) EPCs have been shown to express similar levels of *F8* gene mRNA as cultured hLSECs. Moreover, retroviral-FVIII-transduced UCT-EPCs have proved to be highly amenable to expressing vector-encoded FVIII, producing levels of mRNA, protein, and procoagulant activity that far exceeded those of transduced LSECs, which are thought to be the body’s primary site of FVIII synthesis [203].

Importantly, considering that cord blood is more accessible than liver tissue, these data suggest that cultured vascular-wall-derived UCT EPCs might provide a cellular platform for autologous perinatal FVIII delivery for the treatment of hemophilia A. In mice, the presence of endothelial progenitor cells/vascular endothelial stem cells (EPCs/VESCs) has been identified in different organs, including the liver. Although CD157^+^CD200^+^ EPCs/VESCs express limited levels of *F8* gene mRNA, after transplantation of GFP^+^ into FVIII-deficient mice, wild-type EPCs/VESCs generated liver endothelial cells capable of expressing high levels of *F8* gene mRNA and, consequently, of rescuing the bleeding hemophilic phenotype [204].

These data suggest that endothelial cells may acquire the potential to produce FVIII as they differentiate into specific organ-associated cells. The fact that isolated uncultured human EPCs produce almost no *F8* mRNA [201] is consistent with the idea that endothelial progenitor/stem cells do not produce FVIII. They only start expressing FVIII upon in vitro activation or by integrating and differentiating in the organs’ vasculature.

### 5.3. Factor V

FV levels in full-term neonates are within the same reference ranges as those in adults [87,103]. Nothing is known as yet about the activity of FV during embryonic development. The specific characteristics of the coagulation system in human neonates are well known and are taken into consideration when administering an anticoagulant or a hemostatic so as to adapt them to individuals at that stage of development [96,205,206]. Little information is available on the dynamics of the coagulation system of the human fetus during intra-uterine life. Some authors have shown small differences regarding the maturation of hemostasis between the hemostatic system of preterm neonates born at or beyond 30 weeks of gestation and that of full-term neonates [207,208].

FV levels are typically higher in preterm and newborn individuals than in fetuses [209], which would seem to support the hypothesis that the maturation of the coagulation system occurs during embryonic development. Encoding of mRNA for the coagulation factors is already present in the hepatocytes of the human embryo between the 5th and 10th gestational weeks and is detectable in the plasma of embryos and fetuses between the 8th and 10th weeks of development [210]. It is known that the plasma concentration of clotting factors undergoes changes across the different gestational stages [96,211]. Reverdieau et al. [212] estimated a 25–30% increase in coagulation factor levels between weeks 19 and 29, rising to 45% in the case of FV. This would appear to support the theory that the coagulation system matures during the embryonic and fetal periods.

A direct relationship has been described between the production of FV and the ensuing generation of thrombin [213]. In addition, a series of immunohistological assays [214] have demonstrated that FV is not only present in the liver, the spleen, or the endothelium but also is a marker of the chorionic villi cytotrophoblast. This could suggest that FV may play a role in the formation of thrombin in response to potential damage to the syncytiotrophoblast. Another potential role of FV is an effect on the thrombotic complications associated with lupus anticoagulant occurring during gestation [215].

Studies by various authors [216,217,218,219] geared toward obtaining FV-deficient models for a homozygous severe phenotype were unsuccessful because the individuals analyzed were not viable. Histologic analyses of mouse embryos mutated for FV showed a higher number of somites (from 12 to 16) and an atrophied and granulated yolk sac with collapsed blood vessels. Moreover, the embryos presented abnormal axial rotations and positional defects, underdeveloped or absent cardiac muscle, and small hemorrhages in the mesenchyme of the cephalic region. The mesenchymal tissue was condensed, with several embryos lacking their posterior region. Some FV-mutant individuals were viable but only for a few months [217]. Other mutagenesis studies for FV have demonstrated the importance of the interaction between FV and TFPI for cerebrovascular and glomeruloid body development during embryonic life [218].

These studies show that although its role in embryonic development remains unclear, FV is essential to ensure the embryo’s viability and correct development. It could be hypothesized that FV could be essential for the activation of protease-activated receptors (PARs). The generation of FV-dependent thrombin and of a subsequent signal transmitted by the PAR thrombin receptor are a critical step in early embryonic development, potentially connected to embryonic and yolk sac vasculogenesis [218,220,221,222,223].

Part of these findings are consistent with the results obtained in our laboratory on the expression of *F5* gene mRNA in various tissues during the different stages of embryonic development (work in progress).

Although vWF, FVIII, and FV do participate in the coagulation process, they are synthetized in different ways in the body’s various cells and tissues. Figure 6 shows the in vivo expression of RNA in different organs and tissues according to the classification provided in the Human Protein Atlas. Endothelial cells are the structures exhibiting the highest levels of vWF and FVIII as opposed to extremely low concentrations of FV. The organs responsible for the highest production of vWF are the brain, female tissues, and muscle tissues; those exhibiting the highest expressions of FVIII are the brain, muscle tissues, and female tissues; and those with the greatest secretion of FV are the liver and the gallbladder, followed by the brain and female tissues. All three factors are present at high concentrations in the brain and female tissues, with values in excess of 135 nTPM, and at low concentrations in ocular tissues.

With regard to endothelial cells, they are the chief producers of vWF, although their production of FVIII is also high (at levels above 135 nTPM). FV is only present in these cells in trace amounts.

## 6. Dysregulation of Homeostasis

Dysregulation of the balance between the different physiological systems of homeostasis usually leads to a temporary alteration, which may, however, turn into a permanent condition [224]. At the level of the endothelium and the coagulation system, this pathological state may take the form of either a *direct* dysregulation resulting from a disruption in the function of one or more coagulation factors because of mutational processes or a biogenetic abnormality or an *indirect* dysregulation due to alterations in the components and molecules responsible for regulating the coagulation cascade, the fibrinolytic process, or the endothelium itself. Alterations in protein-based regulatory components are usually due to mutations; a decrease in the concentration of cofactors, such as vitamin K [225]; or an alteration at the sites where some molecules come into contact with the vascular endothelium.

Alterations in coagulation factors often result in a reduction in the activity of a given factor. However, in some cases, they may lead to an increase in clotting function. This is the particular and virtually unique case of FV Leiden, where alterations tend to be associated with hypercoagulability and an ensuing thrombotic risk [226]. Alterations can involve a dysregulation of inhibitors and, generally, of coagulation factor activators. At any event, the balance between coagulation and fibrinolytic systems may be disrupted at any time with an ensuing dysfunction in hemostasis.

Many alterations leading to dysregulation of hemostasis, such as the coagulopathies derived from dysfunctions in some coagulation factor or vascular conditions associated with the endothelium, are rare or ultra-rare diseases, with a prevalence of 1 in every 2000 individuals in the case of rare diseases and 1 in every 50,000 in the case of ultra-rare ones [227,228]. Over 80% of rare diseases are of genetic origin [229]. About 1500 of them are of autosomal origin, 900 are chromosome X linked, 60 are mitochondrial, and around 50 are chromosome Y linked [227].

### 6.1. Coagulopathies

Hemorrhagic coagulation disorders may be associated with alterations in the coagulation cascade or platelet function, or they may stem from a more general condition, such as hepatic or autoimmune disease. Coagulopathies may be acquired or congenital. Congenital coagulopathies, which are derived from a hereditary alteration in the genes coding for different coagulation proteins, may be autosomal or sex linked, dominant or recessive, and monogenic or polygenic [230,231,232]. Different coagulopathies have been described, including von Willebrand’s disease (deficiency of vWF), hemophilia A (deficiency of FVIII), hemophilia B (deficiency of FIX), FXI deficiency, and several alterations in other clotting factors, such as deficiencies of fibrinogen, prothrombin, FV, FVII, and FX, among others [233].

#### 6.1.1. Von Willebrand’s Disease

von Willebrand’s disease (vWD) is an autosomal (dominant or recessive) hemorrhagic coagulopathy characterized by a deficiency of vWF. The condition involves a dysfunction in platelet aggregation and adhesion and a disrupted turnover of FVIII, whose stability depends on vWF. The incidence of this disease is 1 per 100 individuals, although it much depends on the type of disease present [137,139,234]. In this regard, three types of vWD have been described [235]. Type I, which comprises the majority of cases, is characterized by a quantitative deficiency of vWF that usually results in impaired pro-vWF transport (levels < 20 IU/dl) [236]. Type II involves a mutation-derived functional vWF deficiency. Four subtypes have been identified: subtype 2A, which can be subdivided into 2A1, where there is a decreased affinity of vWF for platelets and the subendothelium caused by a deficiency in high-molecular-weight vWF multimers, and 2A2, characterized by increased proteolytic degradation of vWF [237]; subtype 2B, which involves increased affinity of vWF for platelets, resulting in a depletion of platelets and of vWF itself and increasing the risk of thrombotic episodes [238]; subtype 2M, where the GPIb glycoprotein complex, responsible for platelet aggregation and adhesion to the vascular endothelium [237], is reduced or absent; and subtype 2N, where binding of vWF to FVIII is impaired [239]. Lastly, type III vWD is characterized by a total absence of vWF and by extremely low levels of FVIII (1–9 IU/dl) [240].

When transmitted by autosomal dominant inheritance, the disease appears in heterozygous individuals, with mild symptoms (type I or II vWD). If it is transmitted by recessive inheritance, it appears in homozygous or compound heterozygous individuals and symptoms tend to be more severe (type III vWD) [139]. There is usually a close correlation between the mutations in the gene and the different subtypes of the disease. A total of 1217 mutations of the gene that codes for vWF have been included in the Human Gene Mutation Database, put together by the Institute of Medical Genetics (HGMD) [241]. Type I vWD is characterized by the presence of missense mutations, particularly in the D3 domain [242]. The mutations in types 2A2, 2B, and 2M vWD occur mainly in exon 28, which codes for the A1 and A2 domains of vWF. The mutations corresponding to type 2N vWD are to be found in domains D’, D2, and D3. In type 2A1 vWD, mutations are located in the D1 domain and in the CK domain [235,243]. Type III vWD involves the presence of mainly nonsense or frameshift mutations, although deletions, missense, and splice-site mutations have also been observed, with mutations being distributed throughout the gene [240].

Clinical symptoms depend on the activity of vWF, on the type of vWD present, and on the patient’s age and sex [244]. Children present mainly with epistaxis and hematomas, with the most common symptoms in adults being hematomas and bleeding from minor wounds. Women usually present with heavy menstrual bleeding [139,245,246]. Patients with type III vWD usually develop more severe symptoms, such as hemarthrosis, also resulting from low levels of FVIII, and gastrointestinal bleeding, one of the symptoms of gravest concern [240]. Patients with type I disease are typically asymptomatic, and type II patients tend to develop more severe symptoms as they grow older [247].

Diagnosis of the disease is obtained based on a careful investigation of the patient’s family history and on the presence of hemorrhagic symptoms, mainly in the mucosae. Diagnostic tests comprise vWF antigen assays, ristocetin or GPIb-receptor-mediated vWF-platelet-binding assays, and FVIII analyses [234,248].

#### 6.1.2. Hemophilia A

Hemophilia A is a monogenic X-linked congenital or acquired coagulopathy caused by a mutation in the *F8* gene, which codes for FVIII [249,250]. The incidence of the disease is 1 in every 6000 males, with women being the carriers of the disease [227]. With regard to symptoms, the severity of bleeding and of the different symptoms varies according to the FVIII concentrations present. Three categories have been established: mild hemophilia (5–40% of normal FVIII concentrations), moderate hemophilia (1–5% of normal FVIII concentrations), and severe hemophilia (<1% of normal FVIII concentrations) [251,252,253]. Bleeding episodes include epistaxis, hematomas, breakthrough bleeds from joints (hemarthrosis) and muscles, and bleeds following trauma, accidents, or surgical procedures. The most severe cases can involve bleeding in the lungs, the digestive tract, or the central nervous system [249]. There is always a close correlation between circulating factor levels and the phenotype of the disease.

The disease is diagnosed through standard clotting assays, mainly prothrombin time (PT) and activated partial thromboplastin time (aPTT) measurements. Measurements of FVIII levels are made using chromogenic or coagulometric methods [254]. Genetic and molecular assays are also performed to detect mutations in the gene [255,256].

#### 6.1.3. Factor V Deficiency

Congenital FV deficiency is an ultra-rare disease characterized by a dysfunction in coagulation FV. The incidence of the disease is 1 to 9 individuals per million live births [257,258]. It is an autosomal recessive condition, with over 200 mutations having been described of the *F5* gene [241]. The majority of mutations have been found to occur in exon 13, the longest in the gene, which codes for the factor’s B domain [259,260].

Phenotypic clinical manifestations range from mild to severe. Symptoms usually begin at age 6 and comprise soft-tissue or mucosal bleeding, massive fatal bleeding, profuse (nasal and menstrual) bleeding, bleeding during major or even minor surgical procedures and during dental procedures, hematomas, and, in the most severe cases, gastrointestinal, pulmonary, and intracranial hemorrhage. FV levels are not correlated with the severity of the symptoms [257,261,262]. The phenotype may be mild (>10% of normal values), moderate (1–10% of normal values), or severe (<1% of normal values) [257].

Diagnosis is obtained through an investigation of the patient’s family history, coagulation assays, and genetic testing. Coagulation assays usually include PT and aPTT measurements, in addition to a determination of circulating FV levels. Molecular diagnosis is based on the detection of family-specific mutations, given that FV deficiency is associated with a strong consanguinity component. Whole-genome sequencing is the method of choice to detect these mutations and to determine whether they are pathological [229,257,263].

#### 6.1.4. Combined Factor V and Factor VIII Deficiency

This is a rare, clearly consanguinity-related, hemorrhagic autosomal recessive disorder that was first described in 1954 [264,265]. Identified cases amount to a few hundred and are located in countries of the Mediterranean basin, the Middle East, and Southern Asia, probably due to the prevalence of consanguine marriages in those regions [266].

This disorder is characterized by a simultaneous decrease of between 5% and 30% in FV and FVIII plasma levels and by mild-to-moderate hemorrhagic symptoms. Mutations do not directly affect the genes that code for these two factors but rather the *LMAN1* (lectin mannose binding protein 1) gene or the *MCFD2* (multiple coagulation factor deficiency 2) gene [264].

*LMAN1,* a gene located in chromosome 18 (18q21.32), codes for a transport protein located between the cytoplasmic reticulum and the Golgi complex. Mutations of this gene account for 70% of cases of combined FV and FVIII deficiency. This protein joins MCFD2 to form a complex that is responsible for transporting both FV and FVIII from the endoplasmic reticulum to the Golgi complex, from where they are secreted onto the outer surface of FV- and FVIII-producing cells. This disruption in the biogenesis of FV and FVIII, which share this pathway, gives rise to a deficiency in their levels in plasma. *MCFD2,* located in chromosome 2 (2p21), is a gene that codes for the MCFD2 protein, which joins LMAN1 to form said complex. Mutations of this gene account for 15% of cases of combined FV and FVIII deficiency.

From the phylogenetic point of view, the fact that these two factors share a common pathway may be indicative of the fact that both proteins possess similar characteristics. Both are coded by large genes and share a similar sequence (approximately 40% in the A domain and around 35–43% in the C domain) [267]. The B domain is the one with the fewest similarities and the most poorly preserved one through the course of evolution. The proteins are thus large (over 2000 amino acids each) and possess a similar domain structure, the A1 and A2 domains forming the heavy chain, the B domain exhibiting posttranslational modifications, and the A3, C1, and C2 domains making up the light chain [148]. Similarities also exist in terms of functionality, which is consistent with the phylogenetic evolution relationships hypothesis. Their activation also occurs in a similar way, mostly by mediation of thrombin, which binds to arginine residues located in the B domain in order to obliterate said B domain (Figure 4 and Figure 5) [151,161]. The process is identical in the case of FVa and FVIIIa. Moreover, the inactivation of these factors is based on the cleavage and obliteration of the proteins’ A2 domain by APC (Figure 4 and Figure 5) [153,161]. In addition, there is an interaction between both factors as the presence of FV has been shown to boost FVIIIa’s activation efficacy [153].

### 6.2. Thrombophilias

Thrombophilia is a thrombotic disorder associated with a dysfunction in coagulation factors and in some of their cofactors, such as APC and PS [268,269]. Thrombophilia associated with vWF, FVIII, and FV is the result of an impaired turnover of these proteins, which favors the factors’ synthesis, due to an alteration in natural inactivators, such as ADAMTS13 for vWF [270] (Figure 3) and APC and PS for FVIII (Figure 4) and FV (Figure 5), respectively [268,269]. This results in increased levels of vWF, FVIII, and FV, which may trigger a thrombotic process [271].

It may also be the case that a certain coagulation factor may experience a mutation that induces hypercoagulability. Such is the case of FV Leiden. FV Leiden thrombophilia is an autosomal dominant thrombophilic coagulopathy characterized by the presence of a mutation at the binding site to APC in the *F5* gene, specifically at the residue of arginine 506, a cleavage site for APC (Figure 5) [272,273]. The most common clinical manifestations are venous thromboembolism (deep venous thrombosis and pulmonary embolism) and arterial thromboembolism (myocardial infarction and stroke) [274]. Moreover, FV Leiden thrombophilia is associated with an increased risk of thrombosis during pregnancy and the postpartum period, which leads to higher fetal mortality and morbidity rates [275].

### 6.3. Vascular Pathologies Associated with Coagulation Factors

Many of the molecules involved in the coagulation cascade also play a role in the regulation of angiogenesis and neoangiogenesis through their interaction with the vascular endothelium. Several mouse studies that looked into the inactivation of many of the genes encoding critical components of the clotting process, such as TF; thrombin; the PAR1 thrombin receptor; the G protein, which interacts with PAR1 (Gα13); FVIII; FV; and PROS1 (protein C activator), have identified significant vascular defects [276,277]. Although the mechanisms relating hemostasis with angiogenesis remain unclear, FVIII and FV seem to play an important role.

As modulators of the angiogenesis process, vWF and ADAMTS13 may be associated with vasculogenic disease [278]. Weibel–Palade bodies act as reservoirs for vWF and FVIII, as well as for other products related to angiogenesis, such as angiopoietin-2 (Ang-2) [278,279]. vWF may reduce the migration and proliferation of those endothelial cells that depend on vascular endothelial growth factor receptor-2 (VEGFR-2) by inhibiting the release of Ang-2 and raising the levels of integrin αvβ3 [278,279]. Starke et al. [279] observed that vWF-deficient mice exhibit increased angiogenesis and a larger vascular network. In addition, up to 10% of vWD cases (mainly those with type 2A and type 3) develop digestive angiodysplasia, one of the most severe symptoms of the disease [244]. The absence of vWF in endothelial cells leads to greater migration and proliferation (angiogenesis) in response to the VEGF, one of the chief regulators of angiogenesis [279,280].

Hemophilic angiopathy that has been observed in patients with hemophilia A is related to relapsing joint bleeds, inflammation, and the development of new blood vessels. It has been shown that joint bleeding in FVIII-deficient mice induces temporary inflammation, neovascularization, and vessel permeability, followed by vascular remodeling and tissue repair processes, which cannot be fully prevented by short-term FVIII replacement, suggesting that the continuous availability of FVIII plays an important role. No similar phenomena were observed in mice with osteoarthritis or rheumatoid arthritis, suggesting that it is specifically linked to FVIII deficiency. Similar observations have been reported in patients with hemophilia [281,282], where neoangiogenesis/altered vascular architecture was observed in response to bleeding events in the joints, prolonged local inflammation, or vascular permeability. Markers of vascular remodeling, including alpha-smooth muscle actin, endoglin, and VEFG, were strongly expressed after joint bleeding events and were postulated as targets of FVIII signaling [184,282,283].

Although it is not entirely clear what mechanisms are mediated by FVIII, protection against joint-bleeding-induced pathology in hemophilic mice has been achieved by blocking EPCR, a key protein in the activated protein-C-mediated anticoagulant pathway [284,285]. FVIIa binding to EPCR induces PAR1-mediated cell signaling. FVII-EPCR-PAR1-induced signaling was shown to have an anti-inflammatory effect and to offer protection against VEGF-induced endothelial permeability [286]. Further work based on these findings has shown that that EPCR deficiency in FVIII^-/-^ mice significantly reduces the severity of hemophilic synovitis. EPCR deficiency has been shown to attenuate the production of interleukin-6, macrophage infiltration, and neoangiogenesis in the synovium following hemarthrosis. Moreover, administration of a single dose of EPCR-blocking monoclonal antibodies markedly reduced hemophilic synovitis in FVIII^-/-^ mice experiencing joint bleeds. EPCR could therefore constitute an attractive target for preventing joint damage in FVIII-deficient hemophilic patients.

The role of FV in angiogenesis and in vasculogenic disease is indirect. FV is an FXa cofactor of the prothrombinase complex that converts prothrombin into thrombin. Thrombin is perhaps one of the most potent activators of angiogenesis and fulfils this role independently of its fibrin formation function [287]. Thrombin regulates angiogenesis through regulation of multiple pathways, including VEGF, αvb3 integrin, reactive oxygen species, and the expression of the HIF1 signaling pathway. Given that FV participates in thrombin generation, it has been proposed that FV might also be involved in angiogenesis. To demonstrate this point, Yang et al. [288] developed mice with different levels of FV production and found that recovery from hind limb ischemic damage correlated with FV levels. Blood flow, capillary density, and endothelial cell/muscle fiber density were seen to improve in mice presenting with higher levels of FV. Endothelial cell migration assays also showed that FV influences migration and that its activity is mediated by thrombin. Indeed, treatment with a thrombin blocker was found to inhibit endothelial cell migration. Yang et al. also showed that platelet-derived FV has a higher impact on recovery from ischemic damage than circulating FV synthetized in the liver, suggesting that platelet-derived FV contributes to the control of angiogenesis through the control of thrombin generation [288].

Disseminated intravascular coagulation (DIC) is an occlusive process of the circulatory system that affects microcirculation. It occurs mainly as a result of the activation of the coagulation mechanisms that result in clot formation. Bleeding is another common clinical manifestation of DIC, which arises because of an acceleration of fibrinolysis and the consumption of platelets and clotting factors. The combination of thrombosis and hemorrhage gives rise to a severe dysfunction of vital organs resulting from a lack of oxygen supply and the hemorrhage itself. The onset of DIC has been attributed to a systemic inflammatory activation secondary to an underlying condition, which leads to the release of inflammatory mediators, such as tumor necrosis factor (TNF), alpha-tumor necrosis factor (⍺TNF), interleukin 6 (IL-6), and interleukin 1 (IL-1) [70,87,110].

## 7. Homeostasis-Modifying Treatments in Coagulation

Dysregulation of the coagulation system results in pathological alterations, such as coagulopathies or conditions affecting the vascular endothelium. This hemostatic imbalance may be due to a deficiency in coagulation factors vWF, FVIII, and FV; to a hyperactivated coagulation system as a result of a mutation in some factor, such as FV Leiden; or to alterations in the mechanisms involved in activating or inactivating certain coagulation factors.

Consequently, treatments must necessarily be based on reestablishing the lost homeostasis in the hemostatic process. The current understanding of the different pathways and their components, as well as of the relevant molecular and analytical techniques, makes it possible to design multiple strategies based on the various targets on which action is required.

### 7.1. Alteration of the Functionality of Coagulation Factors

Alterations in the levels of a given clotting factor, such as vWF, FVIII, or FV, which are the subject of this article, must be attributed to genetic causes associated with mutations in the gene that encodes them, to an acquired primary autoimmune condition, or to insufficient concentrations of cofactors, such as vitamin K for vitamin-K-dependent factors. This typically results in decreased coagulative activity.

The most commonly used strategy in these cases (Figure 7) consists in replacing the deficient factor by exogenous administration of a recombinant or plasma-derived factor or resorting to new gene- or cell-based therapies. In addition, nonreplacement therapies can be applied, which consist in the administration of agents that mimic some of the coagulation factors and participate in the coagulation cascade or agents that increasing the synthesis of clotting factors within the endothelium.

#### 7.1.1. von Willebrand’s Disease

In the case of vWD, the treatment is based on maintaining vWF levels by administration of desmopressin as non-replacement therapy or, in more severe cases, by administration of an exogenous factor as replacement therapy [289,290,291]. Desmopressin is used to induce vWF production in patients with type I or type II vWD with mild symptoms by increasing the expression of vWF in endothelial cells. Desmopressin may be administered intravenously, intranasally, or subcutaneously [289,291,292]. Desmopressin is a vasopressor that acts at the level of the V2 receptors in the kidney, leading to increased water retention and the translocation of aquaporin channels. These V2 receptors have been found to reside in endothelial cells and to lead to the generation of AMPc. Moreover, V2 receptors stimulate the exocytosis of Weibel–Palade bodies, which is where vWF is synthetized, augmenting the factor’s production two- or even threefold [197,293,294]. In the case of type II vWD, desmopressin is not indicated, as it may induce thrombosis. In the case of type III vWD, it is ineffective [290]. In addition to desmopressin and factor concentrates, antifibrinolytics, such as tranexamic acid or aminocaproic acid, may be indicated to prevent bleeding episodes [295].

Exogenous administration of vWF is not indicated as a prophylactic measure, except in severe cases [290]. This therapy is most commonly applied in patients with type III vWD and in the majority of those with type II vWD. Exogenous therapies may consist in either high-purity vWF concentrates or in low-purity vWF concentrates containing varying amounts of FVIII [296].

#### 7.1.2. Hemophilia A

First-line treatment for FVIII deficiency consists in factor replacement therapy using plasma-derived or recombinant concentrates [297,298,299,300]. This treatment prevents bleeding episodes as well as the joint deterioration resulting from accumulation of blood in the synovial space, which may cause irreversible cartilage damage. Replacement therapy also prevents fatal hemorrhages and muscle spasticity at the level of the central nervous system and the muscles. In patients with moderate or mild bleeding phenotypes, the treatment may be administered on demand, whereby the factor is administered at the time a bleeding episode occurs; in patients with severe bleeding phenotypes, the treatment must be provided regularly and continuously [299,301].

New treatments are now available whereby next-generation, longer-half-life coagulation factors are being used, which allow enhanced therapeutic efficacy and adherence by requiring less frequent administrations. This has a direct impact on the patients’ quality of life [296,302].

For mild bleeding episodes, or in patients bearing the disease, systemic antifibrinolytics, such as tranexamic acid or β-aminocaproic acid, and fibrin- or collagen-based local hemostatic agents have demonstrated significant effectiveness. Moreover, systemic and local hormone therapy and the use of desmopressin have become the treatments of choice in cases of heavy menstrual bleeding [295,303,304].

The latest nonreplacement therapies for hemophilia A have embraced the use of monoclonal antibodies. One such antibody is emicizumab, a recombinant bispecific agent that mimics FVIII and interacts with FIXa and FX to form the tenase complex [305]. Its advantages include its long half-life and its high bioavailability, which make it possible to obtain a stable hemostatic effect following subcutaneous administration [306].

These new cell and gene therapies, although currently in the development phase (particularly the latter), hold significant promise as curative treatments for hemophilia. Indeed, it is believed that they have the potential to cure the disease, unlike currently available treatments, which can only aspire to reduce the symptoms [307].

Gene therapy can be defined as a transfer of DNA that encodes a protein into a cell (or cells) to correct a deficiency or cure a disease. Gene therapy opens up new horizons in the treatment of different kinds of conditions, ranging from genetic diseases to cancer, including infectious, cardiovascular, hepatic, and neurodegenerative diseases [308]. Gene therapy could provide a cure to patients with hemophilia A, ensuring stable and long-lasting concentrations of the deficient circulating factor. Results have to date been encouraging in terms of the level of expression and the duration achieved, especially when adeno-associated vectors are used. However, given that these therapies have been associated with immunogenicity and hepatotoxicity, leading to higher hepatic enzyme levels, work is still in progress to lay the proper foundations to ensure maximum patient safety [309,310]. The few gene therapy clinical trials currently underway in patients with hemophilia A are looking into in vivo viral-vector-mediated gene transfer, where the most commonly used adeno-associated vectors include valoctocogene roxaparvovec [311,312] and giroctocogene fitelparovec [313], among others [314,315].

#### 7.1.3. Factor V Deficiency

The treatment of choice for FV deficiency is currently based on administration of fresh frozen plasma (FFP) [257,316] and on the use of Octaplas^®^ [317], an intravenous preparation characterized by an optimal, extensively tested quantitative and qualitative combination of coagulation factors inactivated against lipid-enveloped viruses. The product also boasts exceptional batch-to-batch consistency, having been shown to remain at a stable temperature of –18 °C for 4 years. It contains high concentrations of fibrinogen and controlled levels of several coagulation factors, fibrinolytics, and their inhibitors (FV levels ≥ 0.5 IU/mL). This product allows a more optimized dosing strategy than FFP.

As in the context of hemophilia A, antifibrinolytics, such as tranexamic acid or β-aminocaproic acid, can be administered orally or intravenously to control (particularly mucosal) bleeding and prior to invasive surgery in patients with FV deficiency. Patients with heavy menstrual bleeding may be administered estrogen or progesterone replacement therapy [257,316,318].

The development in 2014 of the first recombinant FV (^SUPER^FVa) by Drygalski et al. [319] constituted a watershed in the treatment of FV deficiency. This recombinant FV, which is still in the preclinical trial phase, is an FVa capable of resisting the APC-mediated inactivation resulting from the mutations occurring at the three APC-binding sites (Arg306Gln, Arg506Gln, and Arg679Gln). The resistance induced in this recombinant FV enhances the factor’s specific activity three- or fourfold because of the introduction of a disulfide bond in the A2 and A3 domains. The factor presents with optimal safety, efficacy, and stability; a prolonged half-life; low immunogenicity; an absence of thrombogenic effects; and an optimal pharmacogenetic profile [320], all of which make it an ideal prohemostatic candidate for clinical trials. The resistance of ^SUPER^FVa to APC’s anticoagulant effect means that therapeutic strategies based on this recombinant factor are likely to alter the homeostasis process, allowing a longer-term maintenance of FV plasma levels. ^SUPER^FVa’s resistance against inactivation by APC is also an asset in cases of acute traumatic coagulopathy (ATC), characterized by increased levels of APC and hyperfibrinolysis [321].

### 7.2. Rebalancing Therapies

A series of new pharmacological products have been developed within the framework of the so-called hemostatic rebalancing therapies, aimed at restoring the blood’s homeostasis mainly by inhibiting some of the natural anticoagulant pathways, such as TFPI, antithrombin III (AT-III), and APC (Figure 7). These agents, nowadays in more or less advanced clinical trial phases, many of them in phases 2/3, are variously based on monoclonal antibodies, iRNAs, or protease inhibitors [298]. They are considered adjuvant to replacement therapies in the event of a functional deficiency of a coagulation factor, mainly FVIII or FIX.

Phase 3 clinical trials are currently being conducted to analyze two subcutaneously administered anti-TFPI monoclonal antibodies, concizumab [322] and marstacimab [323]. TEPI is an anticoagulant protein that inhibits FXa, either in a direct (and reversible) way or in an indirect way, by joining FXa to form a complex that interferes with the interaction between TF and FVIIa.

Fitusiran (ALN-AT3) is an iRNA-based subcutaneous therapy currently at the clinical trial stage. This agent, which has so far demonstrated promising results in patients with hemophilia A or B, interferes with the synthesis of AT-III. AT-III is a plasma glycoprotein of the SERPIN family that regulates the proteolytic activity of several intrinsic and extrinsic pathway coagulation factors, particularly FVII, IX, FX, XI, and thrombin. To deploy its activity, its basic D-helix binds to the heparin and heparan sulfate proteoglycans present in vascular endothelial cells [324,325].

SerpinPC, a protease inhibitor of the SERPIN family, inactivates APC by rebalancing the coagulation cascade in patients with coagulopathy [326]. APC is a natural anticoagulant that regulates the homeostatic balance of the coagulation system. Produced in the endothelium by mediation of thrombin and TM, APC inactivates mainly FVIII and FV (Figure 4 and Figure 5). Phase 1 and 2 clinical trials [327] have shown that serpinPC is capable of effectively and safely addressing various coagulopathies, particularly hemophilia A and FV deficiency, regardless of their degree of severity. As it acts at the level of APC, it can also be used to treat other kinds of hemorrhagic disorder.

### 7.3. Thrombophilias

In the case of thrombophilia, the dysfunction of homeostasis is associated with an increased rate of thrombosis. The causes may be related to a deficiency of natural anticoagulants, such as PC [268] or PS [269], which tends to result in excessive activity levels of coagulation factors, such as FVIII or FV, or to a state of hypercoagulability typically induced by a mutation in FV Leiden [226], which offers resistance to APC. It may also be due to a deficiency of the regulators of other clotting factors secreted at the level of the vascular endothelium, such as vWF. In this case, a deficiency in vWF’s specific turnover regulator ADAMTS13 [270] may unleash a thrombotic event in patients with thrombotic thrombocytopenic purpura (TTP).

The development of antithrombotic or anticoagulant strategies is an important priority, given the negligible hemorrhagic risk associated with them. However, given that endothelial cells originating in the various vascular sites have different coagulant properties, the hemorrhagic risk can be highly variable [17,23].

#### 7.3.1. Factor V Leiden Thrombophilia

As stated above, FV Leiden thrombophilia is due to a series of mutations in the *F5* gene that result in hypercoagulability of the blood [226]. This is virtually the only instance where an alteration in a coagulation factor destabilizes homeostasis, resulting in hyperfunction. Rebalancing strategies in acute cases are based on the reduction of thrombotic symptoms by using direct oral anticoagulants (DOACs) and on the secondary prevention of venous thromboembolism (VTE) [328,329]. Warfarin is also commonly used in clinical practice as it inhibits gamma carboxylation of precursor proteins, interfering with the synthesis of vitamin-K-dependent coagulation factors [330].

#### 7.3.2. PC Deficiency

PC deficiency drastically alters the homeostasis between procoagulant and anticoagulant proteins, which predisposes subjects to thromboembolism [268]. The role of natural anticoagulants is essential to avoid the consequences of long-term exposure of procoagulant proteins as a result of the low rate of blood flow in the venous circulation. This partly explains the increased incidence of VTE, i.e., deep venous thrombosis (DVT) and pulmonary embolism in young children with PC deficiency. In most cases, treatment is based on the use of anticoagulants, such as heparin, warfarin, aspirin, and clopidogrel [268].

#### 7.3.3. PS Deficiency

PS deficiency is a rare autosomal dominant coagulation disorder characterized by thrombotic events, such as VTE [269]. Treatment is based on administration of anticoagulants following acute VTE episodes. Such agents comprise heparin, vitamin K antagonists, and oral anticoagulants, such as warfarin. The treatment tends to be more prolonged in the case of congenital deficiencies as patients usually take longer to recover from their thrombotic events [269].

#### 7.3.4. Thrombotic Thrombocytopenic Purpura

TTP or Upshaw–Schulman syndrome is a rare autosomal recessive disorder characterized by a deficiency of ADAMTS13. The most usual symptoms are fever, microangiopathic hemolytic anemia, thrombocytopenia, neurological alterations, and renal failure. The disease may result from a mutation in the ADAMTS13 gene or from an autoimmune condition [331]. This deficiency has also been associated with an abnormal distribution of vWF multimers in patients with COVID-19 [332], possibly resulting from increased levels of the proinflammatory cytokine interleukin-6, which are believed to inhibit ADAMTS13′s proteolytic activity [333].

A deficiency of ADAMTS13 leads to the accumulation of platelets and vWF multimers, which results in the formation of micro-clots and in the development of disseminated microvascular ischemia. Under normal physiological conditions, ADAMTS13 cleaves the larger vWF multimers secreted by the Weibel–Palade bodies of endothelial cells [146]. This means that a deficiency of this vWF regulator increases vWF plasma levels, with significant thrombotic effects.

Several therapeutic targets have been proposed to restore homeostatic balance at this level. They include replacement therapies consisting in infusions of plasma or recombinant ADAMTS13, indicated for congenital deficiencies (currently at the phase 3 clinical stage) [334], or the administration of immunomodulators, such as prednisone or rituximab, if the origin of the disorder is autoimmune [335]. In addition, immunotherapy can be administered through anti-vWF monoclonal antibodies, such as caplacizumab [335], a bivalent humanized agent that acts on vWF’s A1 domain, inhibiting its interaction with platelets and preventing high-molecular-weight vWF monomer-mediated platelet adhesion. This strategy attenuates the availability of vWF, which, in turn, leads to transient reductions in the total concentration of the vWF antigen and to a concomitant decrease in FVIII levels during treatment.

#### 7.3.5. Disseminated Intravascular Coagulation

Current therapeutic alternatives for DIC comprise the use of all-trans-retinoic acid (ATRA), combined with anti-fibrinolytic agents, such as tranexamic acid. Recombinant soluble human TM is the treatment of choice for DIC caused by malignant hematologic disease or sepsis [336].

## 8. Conclusions

Due to the high intra- and inter-species heterogeneity of its cells, the vascular endothelium is capable of performing a wide variety of functions. Our increasing understanding of its different functions has revealed that the vascular endothelium plays an important role in hemostasis. It serves as a “meeting point” for multiple coagulation factors and their complexes, made up of both procoagulant and anticoagulant proteins. It also synthetizes important clotting factors, such as vWF.

The homeostasis of hemostasis is based on the balance that exists between procoagulant and anticoagulant and between fibrinolytic and antifibrinolytic proteins, which must be maintained from the embryonic stage to adulthood. *Development hemostasis* refers to the evolution in the maturation of the hemostatic system, whereby every component in the process must adapt its function and its levels to different situations, striving to maintain the hemostatic balance.

If the relationship between structure and function is always an important consideration in physiology, it is particularly significant in the context of hemostasis, as the activation of a given coagulation factor by proteolytic cleavage can only occur if its structure harbors sensitive and specific sites where certain proteases may perform their function. Other sequences are also required for binding the molecules responsible for modulating the activation or inactivation of coagulation factors.

Coagulopathies, which result from a dysregulation of homeostasis, are characterized by a mutation-induced alteration in the functionality of certain coagulation factors or by a dysfunction of the regulators of coagulation or fibrinolysis or of the endothelium itself.

Homeostatic therapies (Table 1) are replacement therapies intended to restore the physiological levels of one or more coagulation factors in the event of a deficiency by infusing recombinant factors, fresh frozen plasma, or Octaplas^®^ or, in the future, administering gene therapy based on an infusion of adeno-associated viruses. Additionally, nonreplacement therapies can be used to simulate the activity of some coagulation factor, such as emicizumab that mimics FVIII or desmopressin that stimulates the production of vWF or FVIII in the endothelium.

The new so-called rebalancing agents, aimed at restoring the homeostasis of hemostasis by inhibiting natural anticoagulant pathways, comprise monoclonal antibodies, iRNAs, and protease inhibitors. Thus, concizumab and marstacimab neutralize TFPI, fitusiran interferes with the synthesis of AT-III, and serpinPC inhibits APC.

Thrombophilias caused by PC or PS deficiencies or by FV Leiden are usually treated with DOACs in regimens carefully tailored to patients’ individual characteristics. In the case of Upshaw–Schulman syndrome, or TTP, which consists in a congenital deficiency of ADAMTS13, a proteolytic regulator of vWF levels, plasma or recombinant ADAMTS13-based replacement therapy has been suggested as an appropriate treatment. For cases where the ADAMTS13 deficiency is caused by an autoimmune mechanism, the treatment of choice is the use of immunomodulators, such as prednisone or rituximab.

Immunotherapy with anti-vWF monoclonal antibodies (caplacizumab) is used in severe cases regardless of whether the cause is congenital or autoimmune, as the purpose of the treatment is to bring down the patients’ elevated vWF levels to prevent the appearance of thrombosis.

Antifibrinolytics agents and recombinant soluble human TM are the standard of care for patients with DIC.

## Figures and Tables

**Figure 1 ijms-23-08283-f001:**
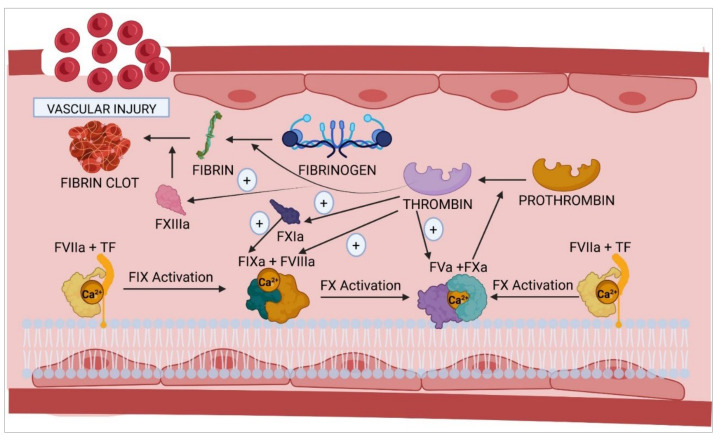
Activation of the coagulation cascade following vascular damage. The initiation complex, formed by the interaction of FVIIa and TF with the membrane phospholipids of the cells in the vascular endothelium, collaborates with calcium to activate FIX and FX to give rise, respectively, to the corresponding tenase and prothrombinase complexes. F, coagulation factor; TF, tissue factor; (+), activation.

**Figure 2 ijms-23-08283-f002:**
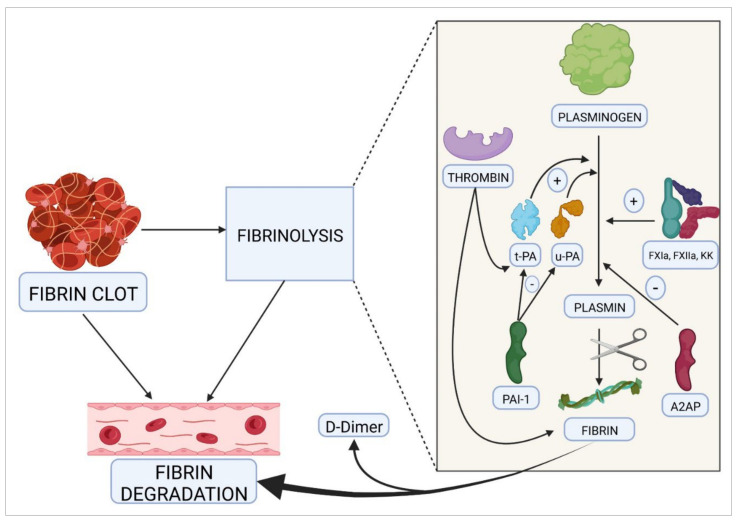
The fibrinolytic process. This pathway is modulated by proteases and clotting factors that stimulate the transformation of plasminogen into plasmin, which is in turn responsible for the subsequent degradation of fibrin. t-PA, tissue plasminogen activator; u-PA, urokinase; PAI-1, plasminogen activator inhibitor-1; A2AP, α2-antiplasmin; FXIa, activated factor XI; FXIIa, activated factor XII; KK, kallikrein; (+), activation; (–), inhibition.

**Figure 3 ijms-23-08283-f003:**
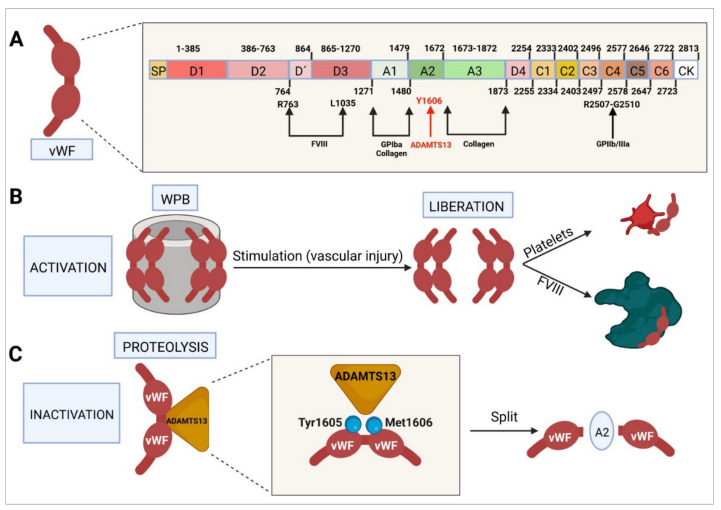
Molecular characteristics of von Willebrand factor. (A) Domain-based structure of the protein, sites at which it interacts with other factors and cofactors, and activation and inactivation sites. (B) WPB-mediated activation and final interaction of vWF with platelets and FVIII. (C) Proteolytic inactivation resulting from interaction with ADAMTS13 and cleavage at the level of the A2 domain. vWF, von Willebrand factor; WPB, Weibel–Palade body; FVIII, factor VIII.

**Figure 4 ijms-23-08283-f004:**
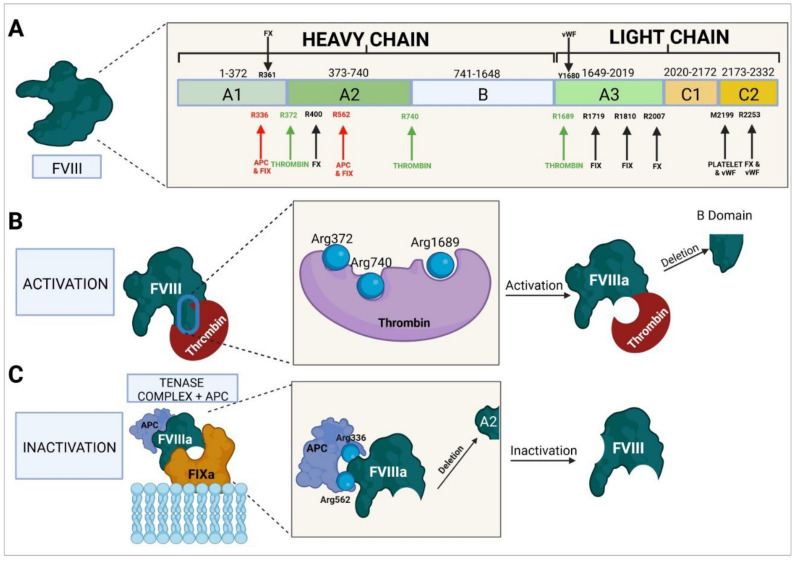
Molecular characteristics of factor VIII. (**A**) Domain-based structure of the protein, sites at which it interacts with other factors and cofactors, and activation and inactivation sites. (**B**) Thrombin-mediated activation and B domain deletion. (**C**) APC-mediated inactivation and A2 domain deletion. FVIII, factor VIII; APC, activated protein C; FIXa, activated factor IX; FX, factor X; vWF, von Willebrand factor.

**Figure 5 ijms-23-08283-f005:**
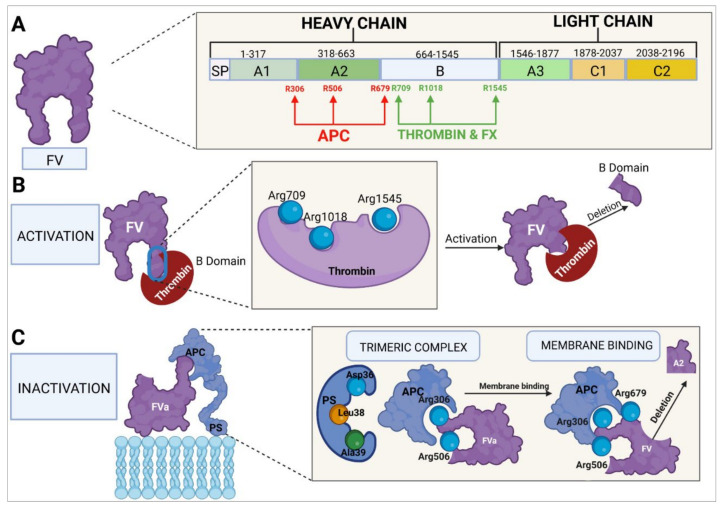
Molecular characteristics of factor V. (**A**) Domain-based structure of the protein, sites at which it interacts with other factors and cofactors, and activation and inactivation sites. (**B**) Thrombin-mediated activation and B domain deletion. (**C**) Inactivation by APC and PS, in conjunction with phospholipids on the membrane of vascular endothelial cells or platelets; deletion of the A2 domain. FV, factor V; FX, factor X; APC, activated protein C; PS, protein S.

**Figure 6 ijms-23-08283-f006:**
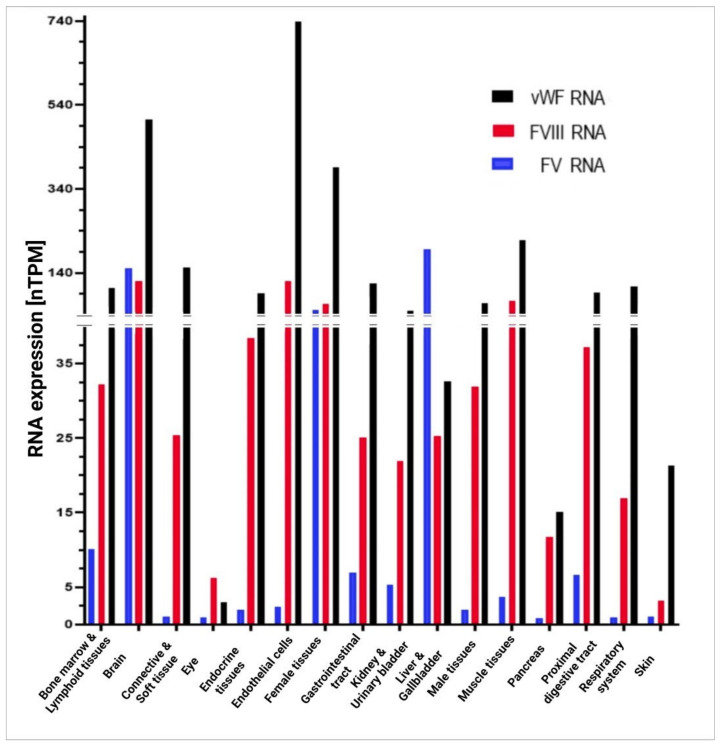
In vivo RNA expression for vWF, FVIII, and FV in different organs and tissues, according to the classification provided by the Human Protein Atlas. Tissues were grouped into organs, such as the bone marrow and lymphoid tissues (appendix, lymph node, spleen, thymus, tonsil), the brain (amygdala, basal ganglia, thalamus, midbrain, pons, medulla oblongata, hippocampal formation, spinal cord, white matter, cerebral cortex, cerebellum, choroid plexus, hypothalamus), connective and soft tissue (soft tissue, adipose tissue), the eye (retina), endocrine tissues (thyroid gland, parathyroid gland, adrenal gland, pituitary gland), female tissues (vagina, breast, cervix, endometrium, fallopian tube, ovary, placenta), the gastrointestinal tract (stomach, colon, duodenum, rectum, small intestine), kidneys and the urinary bladder, the liver and the gallbladder, male tissues (testis, epididymis, prostate, seminal vesicle), muscle tissues (heart muscle, skeletal muscle, smooth muscle), the pancreas, the proximal digestive tract (oral mucosa, salivary gland, esophagus, tongue), the respiratory system (nasopharynx, bronchus, lung), and the skin. nTPM, normalized protein-coding transcripts per million; vWF, von Willebrand factor; FVIII, factor VIII; FV, factor V. Created with GraphPad Prism 8 software (GraphPad Software, La Jolla, CA, USA).

**Figure 7 ijms-23-08283-f007:**
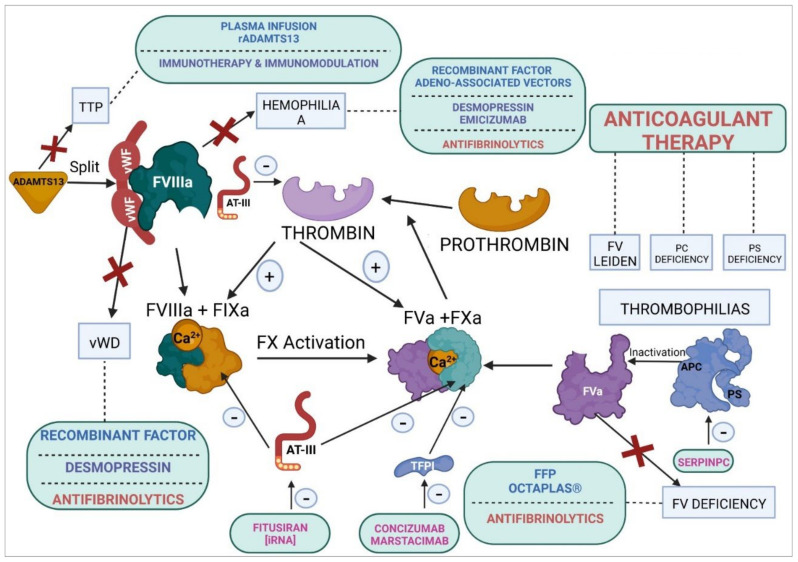
Homeostasis-modifying treatments in hemostasis. Replacement therapies: recombinant factors, fresh frozen plasma, Octaplas^®^, recombinant ADAMTS13, and adeno-associated viruses. Nonreplacement therapies: desmopressin; emicizumab; caplacizumab (anti-vWF); and immunomodulators, such as prednisone and rituximab. Rebalancing therapies: fitusiran, concizumab, marstacimab, and serpinPC. Anticoagulant therapies: DOACs. Antifibrinolytic therapies: tranexamic acid and β-aminocaproic acid. vWF, von Willebrand factor; FVIII, factor VIII; FV, factor V; FIX, factor IX; FX, factor X; TTP, thrombotic thrombocytopenic purpura; vWD, von Willebrand’s disease; FFP, fresh frozen plasma; AT-III, antithrombin III; TFPI, tissue factor pathway inhibitor; APC (PC), activated protein C; PS, protein S; DOACs, direct oral anticoagulants; rADAMTS13, recombinant ADAMTS13. (+), activation; (–), inhibition.

**Table 1 ijms-23-08283-t001:** Homeostasis-modifying treatments: therapeutic indications.

*Drug*	*Pathological Condition*
Desmopressin Recombinant factors Tranexamic acid Aminocaproic acid	von Willebrand’s disease
Recombinant factors (half-life coagulation factors) Emicizumab Tranexamic acid Aminocaproic acid Desmopressin Valoctocogene roxaparvovec/giroctocogene fitelparovec (adeno-associated vectors) ^a^	Hemophilia A
^SUPER^FVa ^b^ Octaplas^®^ Fresh frozen plasma Tranexamic acid Aminocaproic acid Estrogen Progesterone	FV deficiency
Concizumab (anti-TFPI) ^a^ Marstacimab (anti-TFPI) ^a^ Fitusiran (anti-AT-III) ^a^	Hemophilia A or B Other clotting disorders
SerpinPC (APC inhibitor) ^a^	Hemophilia A and FV deficiency Other clotting disorders
DOACs Warfarin	Factor V Leiden thrombophilia
Heparin Warfarin Aspirin Clopidogrel	Protein C deficiency
Heparin Vitamin K antagonists Warfarin	Protein S deficiency
Fresh frozen plasma Prednisone Rituximab rADAMTS13 ^a^ Caplacizumab (anti-vWF)	Thrombotic thrombocytopenic purpura
Trans retinoic acid Tranexamic acid Recombinant human thrombomodulin ^c^	Disseminated intravascular coagulation

^a^ Under investigation in clinical trials. ^b^ Under investigation in pre-clinical trials. ^c^ Only registered in Japan for patients with malignant disease or sepsis. Abbreviations: FVa, activated factor V; TFPI, tissue factor pathway inhibitor; AT-III, antithrombin III; APC, activated protein C; DOACs, direct oral anticoagulants; vWF, von Willebrand factor.

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
