# Peer review of "The Vascular Endothelium and Coagulation: Homeostasis, Disease, and Treatment, with a Focus on the Von Willebrand Factor and Factors VIII and V"

_ijms, 2022, doi:10.3390/ijms23158283_

Round 1

Reviewer 1 Report

I would like to sincerely thank the authors for the comprehensive review with taking into account a relatively high number of evaluated parameters. Please, do not consider the following questins badly - I only suppose that the answers to them may be interesting for the further evaluation of the data from the manuscript:

  1. Can the authors add the similar characteristics of thrombomodulin associated with the system of FV, protein S and protein C ?

  2. May they add the use of described therapeutic advances in the management of disseminated intravascular coagulation ?

  3. From the practical point of view, can the authors make a summary of the current therapeutical indications of described drugs ?

From my perspective, the article can be published after incorporation of the answers to the questions and after such a minor revision.

Author Response

Dear Editor,

I am enclosing a point-by-point response to the referees’ comments. Changes to the text of the revised manuscript have been marked in red and highlighted in yellow using the “track changes” function.

Point-by-point answers in this document are also marked in red.

Reviewer #1 comments

I would like to sincerely thank the authors for the comprehensive review with considering a relatively high number of evaluated parameters. Please, do not consider the following questions badly - I only suppose that the answers to them may be interesting for the further evaluation of the data from the manuscript:

We would like to thank the reviewer for their comments on our manuscript. We believe that constructive criticism is invaluable for the furthering of science.

Can the authors add the similar characteristics of thrombomodulin associated with the system of FV, protein S and protein C?

We have added [lines 167-170 and 342-345] a description of the role of thrombomodulin and its relationship with FV,  PS and PC.

May they add the use of described therapeutic advances in the management of disseminated intravascular coagulation?

At the end of section 6.3. (Vascular pathologies associated with coagulation factors) [lines 1010-1019], we have included, some additional information on disseminated intravascular coagulation. In addition, section 7 (Homeostasis-modifying treatments in coagulation) now contains a new subsection 7.3.5. titled Disseminated intravascular coagulation [lines 1244-1248], which covers the therapeutic advances made with respect to this condition. We have also amended the Abstract [lines 33-34] and added a phrase about this condition in the conclusions section [lines 1292-1293].

As a result of these changes, it was necessary to add a new reference (#336) [line 2241].

From the practical point of view, can the authors make a summary of the current therapeutical indications of described drugs?

Following the reviewer’s suggestion, we have introduced a summary table with the current therapeutic indications of the drugs described in this article (Table 1) [line 2305].

Reviewer 2 Report

This is an extensive review of coagulation, its dysfunctions and regulations, very well documented and described. It tends to provide a global description of hemostasis (less for fibrinolysis), its role and function its regulation and dysfunctions in diseases. This article is very dense and sometimes a bit hard to follow, even for expert persons in the field. It provides anyhow a very complete information on that biological system, as well as an analysis of the rationales for the various activities and their regulation. Globally, this review is pleasant to read, informative and comprehensive. It should have been positively completed by discussing, although brievly, the roles of many hemostasis and fibrinolyis proteins beyond their known primary role. Present understanding shows evidence on the intimate networking of the various biological systems: hemostais, a major body's defense mechanism, is closely interconnected with immunology and inflammation , other major defense mechanisms. Some parts can be condensed and focused better on the major physiological or pathogenic roles of the presented mechanisms.

Concerning the minor comments:
The title is a bit misinforming, as most of the review is on the hemostasis system, vascular endothelium; the evolution from embryo to adulthood is only discussed with the global mechanisms, but is not the review objective by itself.
In paragraph 1.3, endothelial markers are better measured on plasma than on serum to avoid their modification during blood clotting and the generation of enzyme activities.
The endothelial markers cited are not specific for endothelium for most of them, and the authors report a list of all those which are potentially released. I suggest to focus first on those specific for endothelial activity,like vWF and Thrombomodulin or its soluble form (vWF is more a marker of endothelial cell activation, whilst soluble TM is generated through endothelial dammage).
It should be better to name "microparticles", "Extracellular Vesicles (EVs)", which is the term now recommended and which is currently being used. EVs size range from about 0.030 µM to about 5 µM, which is < 0.1 µM to > 1.0 µM. Eventhough some EVs are released by endothelial cells, blood cells or surrounding tissues, the vast majority of EVs in blood circulation are generated from platelets, and are procoagulant.
In paragraph 1.4 TM is exposed on the endothelial cell surface, and binds thrombin, changing its coagulant activity to anticoagulant. The complex is then internalized and TM is recycled later on the endothelial cell surface, and is able to bind again thrombin.
In chapter 2, platelets are involved in the white thrombus, but beyond this characteristic, they have a major role for promoting and growing any blood clot. The white thrombus is only one of the aspects of platelet activity, mainly in arterial thrombus formation .
The description of the coagulation cascade lacks accuracy, respectively to the present understandings. The proeminent blood coagulation mechanism  is the FX activation by TF-FVII (when TF is high) or the FIX activation (when TF is low), the formation of the FIXa-FVIIIa tenase, and subsequently FX activation. The contact phase is more involved in the amplification phase of coagulation.
The description of fibrinolysis is very summarized, and does not show its major role, first in disolving clots in a delayed manner, once the vascular injury is repaired and healing is achieved, then all the other fibrinoysis functions, beyond clot dissolution: in cancer and metastasis, cell remodelling, brain plasticity, fertility, etc. A sheme of the fibrinolysis function, should complete that shown for coagulation.
In paragraph 2.1, please note that among the major hemostasis differences between foetuses/new borns and childhoow/adulthood is the presence of Protein S in the sole free form (no complexes with C4bBP are present).
On page 8, do the authors mean ZPI (Protein Z inhibitor)?
On paragraph 4.2, it should be "gene" and not "gen".
In chapter 6.1, the authors report the FXI deficiency as hemophilia C. This terminology is no more used, due to the different biology and genetics of this deficiency as compared with hemophilias A and B; This term was used a long time ago, but is no more in force.

Author Response

Dear Editor,

I am enclosing a point-by-point response to the referees’ comments. Changes to the text of the revised manuscript have been marked in red and highlighted in yellow using the “track changes” function.

Point-by-point answers in this document are also marked in red.

Reviewer #2 comments:

This is an extensive review of coagulation, its dysfunctions, and regulations, very well documented and described. It tends to provide a global description of hemostasis (less for fibrinolysis), its role and function its regulation and dysfunctions in diseases. This article is very dense and sometimes a bit hard to follow, even for expert persons in the field. It provides anyhow a very complete information on that biological system, as well as an analysis of the rationales for the various activities and their regulation. Globally, this review is pleasant to read, informative and comprehensive. It should have been positively completed by discussing, although briefly, the roles of many hemostasis and fibrinolysis proteins beyond their known primary role. Present understanding shows evidence on the intimate networking of the various biological systems: hemostasis, a major body's defense mechanism, is closely interconnected with immunology and inflammation, other major defense mechanisms. Some parts can be condensed and focused better on the major physiological or pathogenic roles of the presented mechanisms.

We are grateful to the reviewer for their general comments on our manuscript. To address their comment that some parts (of the article) can be condensed and focused better on the major physiological or pathogenic roles of the presented mechanisms, a few paragraphs have been added taking care not to significantly alter the structure or the contents of the paper.

Concerning the minor comments:

The title is a bit misinforming, as most of the review is on the hemostasis system, vascular endothelium; the evolution from embryo to adulthood is only discussed with the global mechanisms but is not the review objective by itself.

We agree with the reviewer’s comments. The title was worded to respond to the theme of the special issue it was written for, but it is true that we were not 100% satisfied with it. As a result, we have now changed the title to a form we hope will be more to the reviewer’s liking [line 3].

In paragraph 1.3, endothelial markers are better measured on plasma than on serum to avoid their modification during blood clotting and the generation of enzyme activities.

We have emphasized that plasma is the fraction where biomarkers are identified (for the reasons mentioned by the reviewer) [line 102].

The endothelial markers cited are not specific for endothelium for most of them, and the authors report a list of all those which are potentially released. I suggest to focus first on those specific for endothelial activity, like vWF and Thrombomodulin or its soluble form (vWF is more a marker of endothelial cell activation, whilst soluble TM is generated through endothelial damage).

The reviewer is absolutely right. We have modified the text accordingly [lines 106-110] making reference  to the specific endothelial markers: vWF and thrombomodulin (in its free or conjugated forms).

It should be better to name "microparticles", "Extracellular Vesicles (EVs)", which is the term now recommended and which is currently being used. EVs size range from about 0.030 µM to about 5 µM, which is < 0.1 µM to > 1.0 µM. Even though some EVs are released by endothelial cells, blood cells or surrounding tissues, the vast majority of EVs in blood circulation are generated from platelets and are procoagulant.

Following the reviewer’s instructions, we have replaced the term “microparticles” by the more correct term “extracellular vesicles” (EVs) throughout the text [lines 145, 147, 151, 153 and 154].

In paragraph 1.4, TM is exposed on the endothelial cell surface, and binds thrombin, changing its coagulant activity to anticoagulant. The complex is then internalized, and TM is recycled later the endothelial cell surface and is able to bind again thrombin.

We have revised paragraph 1.4 in accordance with the reviewer’s suggestion [lines 167-170 and 342-345].

In chapter 2, platelets are involved in the white thrombus, but beyond this characteristic, they have a major role for promoting and growing any blood clot. The white thrombus is only one of the aspects of platelet activity, mainly in arterial thrombus formation.

We have made the relevant changes to section 2 in accordance with the reviewer’s suggestion [lines 202-205].

The description of the coagulation cascade lacks accuracy, respectively to the present understandings. The preeminent blood coagulation mechanism is the FX activation by TF-FVII (when TF is high) or the FIX activation (when TF is low), the formation of the FIXa-FVIIIa tenase, and subsequently FX activation. The contact phase is more involved in the amplification phase of coagulation.

We are thankful to the reviewer for this suggestion. Our goal in this article was to focus on other aspects. For that reason, no in-depth analysis was made of the workings of the coagulation cascade. However, further to the reviewer’s suggestion, we have added a paragraph [lines 217-219 and 228-233] to bring home the importance of the contact phase or extrinsic pathway in the coagulation process.

The description of fibrinolysis is very summarized, and does not show its major role, first in dissolving clots in a delayed manner, once the vascular injury is repaired and healing is achieved, then all the other fibrinolysis functions, beyond clot dissolution: in cancer and metastasis, cell remodeling, brain plasticity, fertility, etc. A scheme of the fibrinolysis function should complete that shown for coagulation.

We thank the reviewer for their interesting remark. We have accordingly elaborated on the information provided in section 2 regarding fibrinolysis [lines 258-273] to emphasize the important role it plays in the final stages of hemostasis. Moreover, we have included a new figure (new Figure 2) [line 281], which provides a graphic explanation of how this pathway works. This made it necessary to change the numbering of all subsequent figures in the text.

In paragraph 2.1, please note that among the major hemostasis differences between foetuses/new burns and childhood/adulthood is the presence of Protein S in the sole free form (no complexes with C4bBP are present).

Following the reviewer’s suggestion, we have added a comment about this aspect and has been referenced (Ref.#80) [lines 304-306].

On page 8, do the authors mean ZPI (Protein Z inhibitor)?

Indeed, ZPI is the complex formed by protein Z (PZ) and its inhibitory protease (PI). This has been clarified in the text [line 338], and ZPI has been placed between brackets.

On paragraph 4.2, it should be "gene" and not "gen".

Thank you, we have corrected the spelling of the word [line 456]

In chapter 6.1, the authors report the FXI deficiency as hemophilia C. This terminology is no more used, due to the different biology and genetics of this deficiency as compared with hemophilias A and B; This term was used a long time ago but is no more in force.

The reviewer is right. We have replaced “hemophilia C” by “FXI deficiency” [lines 808-809].
